



# GloFAS-ERA5 operational global river discharge reanalysis 1979-present

Shaun Harrigan[1], Ervin Zsoter[1], Lorenzo Alfieri[2], Christel Prudhomme[1,3,4] Peter Salamon[2], Fredrik Wetterhall[1], Christopher Barnard[1], Hannah Cloke[5,6,7,8], and Florian Pappenberger[1]

[1]Forecast Department, European Centre for Medium-Range Weather Forecasts (ECMWF), Reading, UK
[2] Disaster Risk Management Unit, European Commission Joint Research Centre (JRC), Ispra, Italy
[3]Centre for Ecology and Hydrology (CEH), Wallingford, UK
[4]Department of Geography and Environment, University of Loughborough, Loughborough, UK
[5]Department of Geography and Environmental Science, University of Reading, Reading, UK
[6]Department of Meteorology, University of Reading, Reading, UK
[7]Department of Earth Sciences, Uppsala University, Uppsala, Sweden
[8]Centre of Natural Hazards and Disaster Science, CNDS, Uppsala, Sweden

*Correspondence to*: Shaun Harrigan (shaun.harrigan@ecmwf.int)

**Abstract.** Estimating how much water is flowing through rivers at the global scale is challenging due to a lack of observations in space and time. A way forward is to optimally combine the global network of earth system observations with advanced numerical weather prediction (NWP) models to generate consistent spatio-temporal maps of land, ocean, and atmospheric variables of interest, known as a reanalysis. While the current generation of NWP output runoff at each grid cell, they currently do not produce river discharge at catchment scales directly, and thus have limited utility in hydrological applications such as flood and drought monitoring and forecasting. This is overcome in the Global Flood Awareness System (GloFAS; http://www.globalfloods.eu/) by coupling surface and sub-surface runoff from the HTESSEL land surface model used within ECMWF's latest global atmospheric reanalysis (ERA5) with the LISFLOOD hydrological and channel routing model. The aim of this paper is to describe and evaluate the GloFAS-ERA5 global river discharge reanalysis dataset launched on 5 November 2019 (version 2.1 release). The river discharge reanalysis is a global gridded dataset with a horizontal resolution of 0.1° at a daily time step. An innovative feature is that it is produced in an operational environment so is available to users from 1 January 1979 until near real time (within 7 days behind real time). The reanalysis was evaluated against a global network of 1801 river discharge observation stations. Results found that the GloFAS-ERA5 reanalysis was skilful against a mean flow benchmark in 86 % of catchments according to the modified Kling-Gupta Efficiency Skill Score, although the strength of skill varied considerably with location. The global median Pearson correlation coefficient was 0.61 with an interquartile range of 0.44 to 0.74. The long-term and operational nature of the GloFAS-ERA5 reanalysis dataset provides a valuable dataset to the user community for applications ranging from monitoring global flood and drought conditions, identification of hydroclimatic variability and change, and as raw input to post-processing and machine learning methods that can add further value. The dataset is openly available from the Copernicus Climate Change Service Climate Data Store:





**1 Introduction**

A key challenge in hydrology is estimating past, present, and future hydrological conditions in rivers around the world. This is largely due to severe temporal and spatial gaps in the global river discharge observing network. In many parts of the world there is simply not enough long-term river discharge observations at high enough spatial density, and in the vast majority of
countries hydrometric data are not available in real time (Lavers et al., 2019). The lack of observations is therefore a major barrier in our ability to provide monitoring and early warning of hydrological extremes such as floods and droughts, which has for example implications for progressing international disaster risk reduction (UNDRR, 2015). A way forward pioneered in the field of meteorology and climate has been to optimally combine in situ and satellite earth system observations together with advanced numerical weather prediction (NWP) models to generate a 'reanalysis' of land, ocean, and atmospheric variables
of interest, thus providing consistent spatio-temporal "maps without gaps" (Hersbach et al., 2018). Several global hydrological products have been developed that provide estimates of runoff or river discharge, with a wide range of forcing and methodological approaches (e.g. Fekete et al., 2002; Döll et al., 2003; Qian et al., 2006; Sperna Weiland et al., 2010; Reichle et al., 2011; Yamazaki et al., 2011; Beck et al., 2017; Ghiggi et al., 2019; Lin et al., 2019). While these datasets can be used to understand past variability and change in the terrestrial hydrological cycle, they are currently not produced in an operational
environment in near real time, so cannot be used for monitoring current global river conditions or provide initial conditions to hydrometeorological forecasting systems.

A long term and near real time river discharge reanalysis is produced operationally as part of the Global Flood Awareness System (GloFAS; http://www.globalfloods.eu/) which bridges this gap. GloFAS is the global flood service of the European
Commission's Copernicus Emergency Management Service (CEMS), an operational system for monitoring and forecasting floods across the world with over 4000 registered users. GloFAS was developed together by the Joint Research Centre (JRC) of the European Commission, the University of Reading, and the European Centre for Medium-Range Weather Forecasts (ECMWF). The system went pre-operational in July 2011 (Alfieri et al., 2013), becoming a fully operational 24/7 supported service in April 2018 (version 1.0, upgraded to version 2.0 in November 2018). GloFAS is provided through a free and open
licence and is designed for decision makers and forecasters in national and international water authorities, water resources management, hydropower companies, civil protection authorities, and international humanitarian aid organisations. A recent example of the use of GloFAS was for supporting the humanitarian response to the devastating floods that affected large parts of Mozambique, Malawi, and Zimbabwe in the wake of tropical cyclone Idai in March 2019 (Magnusson et al., 2019). Given the large amount of openly available data that is generated by GloFAS, including a long-term near real time river discharge



reanalysis, a large set of reforecasts, and real time flood and seasonal forecasts, it is also used by researchers and commercial industries for a wide range of projects and for developing value-added products.

In GloFAS, ensemble river discharge forecasts are produced each day at a daily time step and provide probabilities of exceeding flood thresholds for a given river section with a lead time out to 30 days ahead (GloFAS 30-day; Alfieri et al., 2013).

There is also a seasonal component, GloFAS-Seasonal (Emerton et al., 2018), that provides forecasts once per month at a weekly time step with a lead time out to four months ahead. The river discharge reanalysis is used for two core tasks within GloFAS. First, flood thresholds at 2-, 5-, and 20-year return periods for each river cell are derived from the long-term reanalysis series. This allows for the magnitude of the real time ensemble river discharge forecasts to be directly compared to the magnitude of the long-term flood thresholds, and thus awareness of a flood signal if the threshold is exceeded. Second, it

provides the basis to derive initial hydrometeorological conditions for both GloFAS 30-day and GloFAS-Seasonal real time forecasts. Estimating initial conditions is a key step to determine current status of soil moisture, groundwater, snow cover, and initial state of water within rivers and other waterbodies and has been identified as one of the major challenges in continental and global scale flood forecasting given the limited availability of observational data at these scales (Emerton et al., 2016).

The aim of this data paper is to describe the newly produced operational river discharge reanalysis dataset as part of the launch of GloFAS v2.1 on 5 November 2019 (see GloFAS technical documentation for details on upgrades: https://confluence.ecmwf.int/display/COPSRV/GloFAS). GloFAS river discharge reanalysis is based on ERA5 (Hersbach et al., 2018), ECMWF's latest global atmospheric reanalysis which extends back to 1979, officially released in January 2019. An innovation of ERA5 is that it is produced in near real time in an operational environment, allowing for the production of

GloFAS-ERA5 reanalysis within 7 days of real time. This has the major advantage for GloFAS that the initial hydrometeorological conditions can now be derived from the same product as the long-term flood thresholds are derived, so will ensure much better consistency with real time forecasts compared to previous GloFAS model configurations. Uniquely, the global river discharge product is over 40 years long, produced in near real time, and is freely available to download for the community through the Copernicus Climate Change Service (C3S) Copernicus Climate Data Store (CDS):

https://cds.climate.copernicus.eu/cdsapp#!/dataset/cems-glofas-historical?tab=overview (C3S, 2019), opening multitudes of hydroclimate applications across the world.

Section 2 outlines the production of the dataset and Sect. 3 describes its main attributes including available variables and file format. An evaluation of the dataset against a global network of observations is conducted in Sect. 4. The dissemination of the

data through the CDS is shown in Sect. 5 before key conclusions and future work are offered in Sect. 6.

## 2 Data production

Pappenberger et al. (2010) first demonstrated that it was possible to achieve useful river discharge predictions by coupling a river routing scheme with the land surface model of the ECMWF global numerical weather prediction (NWP) system. The GloFAS-ERA5 river discharge reanalysis uses this concept and is produced by coupling the land surface model runoff component of the ECMWF ERA5 global reanalysis (Hersbach et al., 2018) with the LISFLOOD hydrological and channel routing model (van der Knijff et al., 2010). In ERA5 the runoff (m d$^{-1}$) from one cell is not connected to neighbouring cells, hence it is not possible to estimate river discharge (m$^3$ s$^{-1}$) at the catchment scale. Coupling ERA5 runoff with LISFLOOD allows for lateral connectivity of grid cells with runoff routed through the river channel to produce river discharge. A schematic of the key components in the production of the GloFAS-ERA5 reanalysis is provided in Fig. 1 and described below.

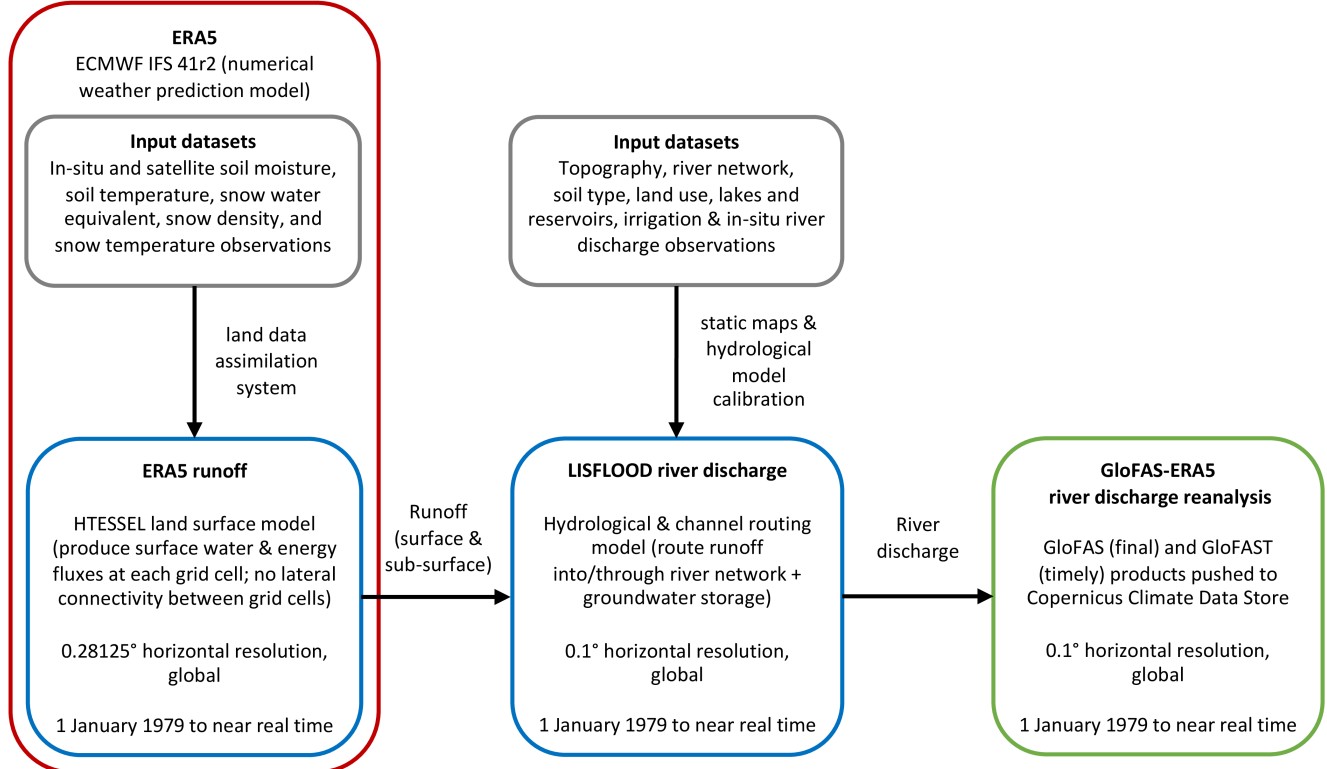

**Figure 1:  A schematic of the key components in the production of GloFAS-ERA5 river discharge reanalysis dataset.**

### 2.1 ERA5 runoff

ERA5 runoff is produced from the HTESSEL land surface model (Hydrology Tiled ECMWF Scheme for Surface Exchanges over Land; Balsamo et al., 2009) as used within the ECMWF Integrated Forecasting System (IFS). HTESSEL computes the surface water and energy fluxes, and the temporal evolution of soil temperature, soil moisture, and snowpack. ERA5 uses an advanced land data assimilation system to assimilate conventional in-situ and satellite observations for land surface variables



such as soil moisture, soil temperature, snow water equivalent, snow density, and snow temperature as outlined in de Rosnay et al., (2014).

ERA5 benefits from a decade worth of numerical weather prediction (NWP) developments in model physics, numerics, and data assimilation by using ECMWF IFS model cycle 41r2 (2016) compared to model cycle 31r2 (2006) as used in its predecessor, ERA-Interim (Dee et al., 2011). ERA5 has a horizontal resolution of approximately 31 km at the equator (native octahedral grid) and since January 2019 is openly available from 1979 to present. A key novelty of ERA5 is its operational production that makes available an intermediate timely product, ERA5T in near real time, allowing the production the GloFAS-
ERA5 river discharge reanalysis operationally within 7 days of real time.

**2.2 LISFLOOD river discharge**

River discharge is currently not calculated by HTESSEL. Instead, surface and sub-surface runoff from the HTESSEL land surface model are coupled with a simplified global version of LISFLOOD, a spatially distributed grid-based hydrological and channel routing model. The details of the global version of LISFLOOD used within GloFAS and its calibration can be found
in Hirpa et al. (2018) but are briefly summarised here for context. The sub-surface runoff from HTESSEL is used as input to the LISFLOOD groundwater module, which consists of two parallel linear reservoirs that store and subsequently transport water to the river channel with a time delay. The upper zone represents quick groundwater and sub-surface flow while the lower zone represents slow groundwater flow that generates base flow. The surface runoff from HTESSEL is used as input to the LISFLOOD river channel routing module. This is a two-stage process whereby the surface runoff for each cell is first
routed to the nearest downstream river channel cell, then the water in the channel is routed through the river network using the kinematic wave approach. Groundwater and river routing parameters in GloFAS were calibrated against river discharge observations for 1287 catchments globally by Hirpa et al. (2018). A key feature of LISFLOOD is the ability to represent features such as lakes and reservoirs that can severely alter the timing and magnitude of river discharge. A total of 463 of the largest lakes (surface area > 100 km$^2$) and 667 largest reservoirs have been incorporated into the GloFAS river network by
Zajac et al. (2017).

To generate the GloFAS-ERA5 river discharge reanalysis, the global LISFLOOD model is forced with daily HTESSEL surface and sub-surface runoff from ERA5 that has been resampled to the 0.1° GloFAS gridded river network starting from 1 January 1979 (Fig. 1). LISFLOOD was given a one-year model spin up using preliminary ERA5 output for 1978. To produce GloFAS-
ERA5 reanalysis in near real time operationally, the latest available ERA5T data is used.



## 3 Data description

The key attributes of the current operational version (v2.1) of the GloFAS-ERA5 river discharge reanalysis dataset are shown in Table 1. The daily reanalysis is global in coverage, except for Antarctica, with a horizontal grid resolution of 0.1° (approximately 11 km at the equator). The dataset is over 40 years long starting 1 January 1979. An innovative aspect of the

dataset is its operational production allowing it to be available within 7 days of real time, shortly after ERA5T becomes available. The intermediate ERA5T data is not quality assured due to its timely nature. Consequently, there will be two reanalysis streams available: GloFAS (consolidated) is the final product based on the consolidated ERA5 from 1 January 1979 to 2 - 3 months behind real time, updated on the CDS on a monthly basis; and GloFAST (intermediate) is the timely product based on the intermediate ERA5T from 1 August 2019 to within 7 days of real time, updated on the CDS on a daily basis

whenever ERA5T becomes available.

The GloFAS-ERA5 reanalysis dataset includes the variables river discharge and the upstream area for each GloFAS grid cell (Table 2). Data are stored in NetCDF format with one file per day containing the 24 h mean river discharge (00 UTC to 00 UTC). Each daily filename follows the convention 'CEMS_ECMWF_dis24_<YYYYMMDD>_glofas<T>_v2.1.nc' whereby

the    date    stamp    represents    the    end    of    the    24 h    averaging    period.    So,    for    example    the    file 'CEMS_ECMWF_dis24_20190101_glofas_v2.1.nc' contains the daily mean flow for the 24 h period 00 UTC 2018-12-31 to 00 UTC 2019-01-01. Appendix A shows the header metadata information contained within the example NetCDF file. Each daily NetCDF file for the whole globe has an uncompressed size of ~21.7 MB, therefore the estimated size of the dataset from January 1979 to October 2019 is ~320 GB.

Figure 2 maps the mean daily river discharge over 1979 to 2018 for each GloFAS river with an upstream area greater than 1000 km$^2$, revealing the main river arteries of the world. An example hydrograph of the long-term near real time reanalysis against available river discharge observations is shown in Fig. 3 for the Teles Pires River in the Amazon basin, Brazil.

**Table 1: Summary of GloFAS-ERA5 dataset attributes on the C3S Climate Data Store**

| Dataset attribute | Details |
|---|---|
| Horizontal coverage | Global except for Antarctica (90° N-60° S, 180° W-180° E) |
| Horizontal resolution | 0.1° x 0.1° |
| Spatial reference system | Latitude/Longitude (WGS 84, EPSG:4326) |
| Vertical resolution | Surface level for river discharge |



| Temporal resolution | Daily data |
|---|---|
| Temporal coverage | 1979-01-01 to near real time |
| Availability behind real time | i.) GloFAS (consolidated): 2 - 3 months, updated on CDS monthly (final product following availability of officially released quality assured ERA5 data)<br>ii.) GloFAST (intermediate): within 7 days of real time, updated on CDS daily (timely product following availability of non-quality assured ERA5T data) |
| Update frequency | A new river discharge reanalysis will be published with every major update of the GloFAS system. The latest version will always be the version used in operations |
| File format | NetCDF |
| Data type | Grid |
| Data size on disk | Approximately 21.7 MB uncompressed per global NetCDF file for one day (full dataset currently ~320 GB uncompressed) |
| Version | GloFAS-ERA5 v2.1 |
| File naming convention | 'CEMS_ECMWF_dis24_<YYYYMMDD>_glofas<T>_v2.1.nc' where YYYY is year, MM is month, DD is day, and T is for timely (i.e. GloFAST). The date stamp, <YYYYMMDD>, represents the end of the 24 h averaging period |

**Table 2: Variables available within GloFAS-ERA5 dataset on the C3S Climate Data Store**

| Variable type | Name | Units | Description |
|---|---|---|---|
| Primary variable | River discharge | $m^3\ s^{-1}$ | Volume rate of water flow, including sediments, chemical and biological material, in the river channel averaged over a time step through a cross-section. The value is an average over a 24 h period |
| Related variable | Upstream area | $m^2$ | Static file ('upArea.nc'), Upstream area for the point in the river network |

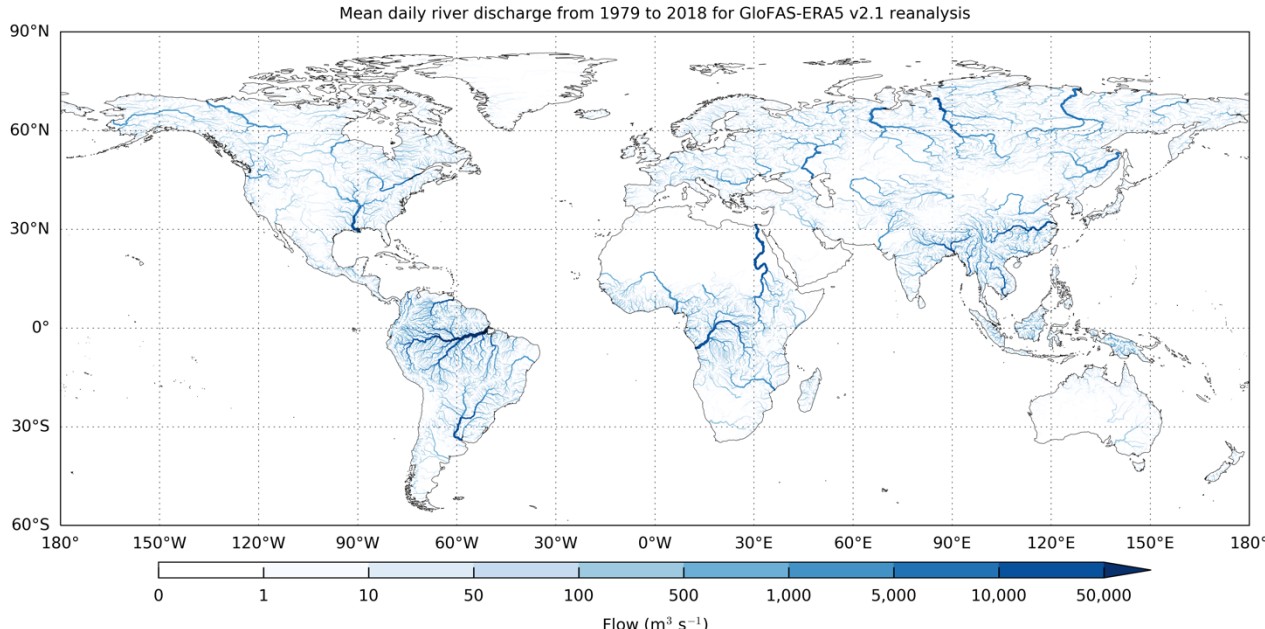

**Figure 2: Mean GloFAS-ERA5 daily river discharge over 1979 to 2018 for each GloFAS river grid cell with an upstream area greater than 1000 km². Darker blue river sections have larger river discharge.**

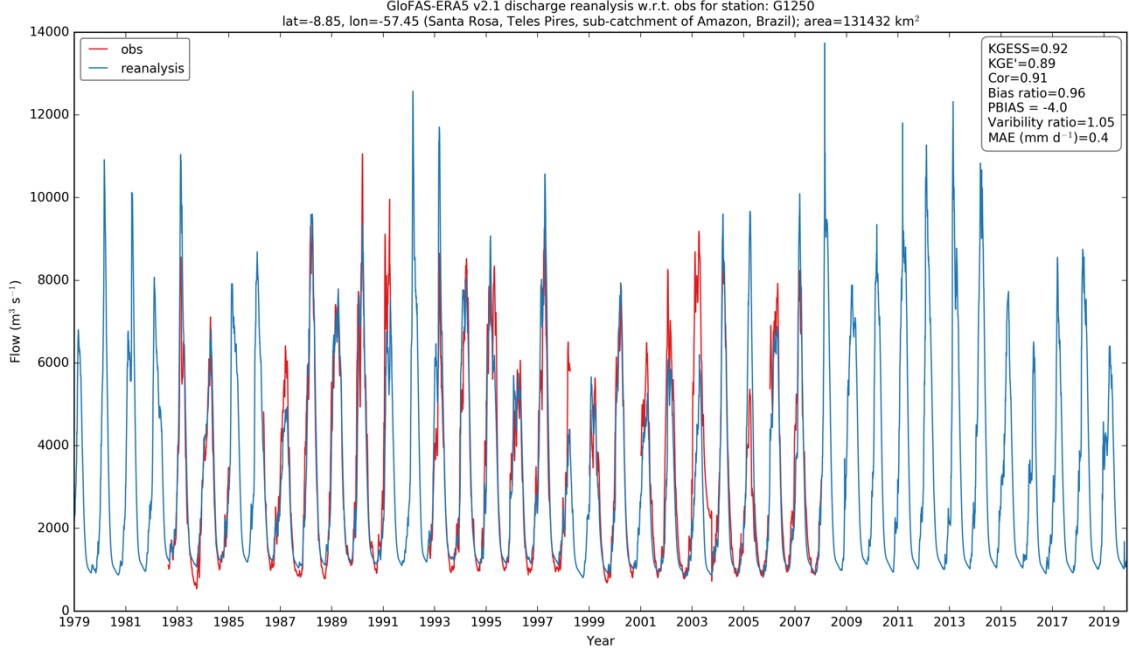

**Figure 3: Hydrograph for GloFAS-ERA5 river discharge reanalysis (blue line) from 1 January 1979 to 12 November 2019 and observations (red line), when available, for the Santa Rosa gauging station on the Teles Pires River, a sub-catchment of the Amazon, Brazil (GloFAS ID=1250; GRDC ID=3629770). Summary statistics from evaluation of the reanalysis against observations in top right box as used in Sect. 4.**



## 4 Evaluation and limitations

GloFAS-ERA5 v2.1 river discharge reanalysis was evaluated against a global network of river discharge observations. As part of GloFAS a database of global hydrological observations for 2042 stations is held, consisting predominantly (i.e. ~75 %) from the Global Runoff Data Centre (GRDC) and supplemented by data collected through collaboration with GloFAS partners worldwide to improve spatial coverage. A number of criteria were used to select the station list:

- At least 4 years of data available between 1979 and 2018 (not necessarily contiguous) [78 stations removed]
- Minimum upstream area of 500 km$^2$ [4 stations removed]
- Error in catchment area supplied by data provider and upstream area for corresponding cell on the GloFAS river network within 20 % [93 stations removed]
- When multiple observation stations were matched to the same GloFAS river cell, only one was selected [27 stations removed]
- First order visual quality check on observed river discharge time-series to remove stations with erroneous data (for example, time series truncated above a threshold, severe inhomogeneities, or series monitoring an artificial canal instead of a river) [39 stations removed]

This filtering procedure resulted in the selection of 1801 catchments with drainage areas ranging between 575 km$^2$ to 4,664,200 km$^2$, and a median of 30,046 km$^2$. Care must be taken in spatial representativeness of the following evaluation results as the observation network is sparse in some regions of the world, particularly in large parts of Africa and Asia.

Performance was assessed using the modified Kling-Gupta Efficiency metric (KGE'; Gupta et al., 2009; Kling et al., 2012). The KGE' is gaining popularity as the standard performance metric in hydrology (e.g. Beck et al., 2017; Harrigan et al., 2018; Lin et al., 2019) and can be decomposed into three components important for assessing hydrological dynamics: temporal errors through correlation, bias errors, and variability errors:

$$KGE' = 1 - \sqrt{(r - 1)^2 + (\beta - 1)^2 + (\gamma - 1)^2} \tag{1}$$

$$\beta = \frac{\mu_s}{\mu_o} \tag{2}$$

$$\gamma = \frac{\sigma_s / \mu_s}{\sigma_o / \mu_o} \tag{3}$$

where $r$ is the Pearson correlation coefficient between reanalysis simulations ($s$) and observations ($o$), $\beta$ is the bias ratio, $\gamma$ is the variability ratio, $\mu$ the mean discharge, and $\sigma$ the discharge standard deviation. The KGE' and its three decomposed components (correlation, bias ratio, and variability ratio) are all dimensionless with an optimum value of 1. The bias ratio $\beta$



can be easily converted into the more commonly used percent bias (PBIAS (%)) by $(\beta - 1) \times 100$. In order to evaluate the hydrological simulation *skill* of GloFAS-ERA5 reanalysis, its performance is compared against a simpler benchmark. Here the observed mean flow is used as a benchmark as proposed by Knoben et al. (2019). This is not a difficult benchmark to beat but should arguably be the minimum reference for any hydrological system to be compared against. Here we represent KGE' as a

skill score, KGESS, to evaluate the performance of GloFAS-ERA5 river discharge reanalysis against the mean flow benchmark simulation, given as:

$$KGESS = \frac{KGE'_{reanalysis} - KGE'_{bench}}{KGE'_{perf} - KGE'_{bench}} \tag{4}$$

where $KGE'_{reanalysis}$ is the KGE' value for the GloFAS-ERA5 reanalysis against observations, $KGE'_{bench}$ is the KGE' value for

the observed mean flow benchmark against observations (i.e. $KGE'(\overline{Q_{obs}}) = 1 - \sqrt{2} \approx -0.41$ from Knoben et al. (2019)), and $KGE'_{perf}$ is the value of KGE' for a perfect simulation which is 1. A KGESS = 0 means the GloFAS-ERA5 reanalysis is no better than the mean flow benchmark so has no skill, KGESS > 0 for when the reanalysis is considered skilful, and KGESS < 0 for when performance is worse than the benchmark so has negative skill.

**4.1 Overall performance**

Results for overall performance show that the GloFAS-ERA5 river discharge reanalysis is skilful in 86 % of catchments (Fig. 4). The global median KGESS is 0.51 with an Interquartile range (IQR) of 0.3 to 0.66. Performance is best in Brazil (particularly the Amazon basin), central Europe, and eastern and western regions of the US. GloFAS-ERA5 reanalysis performance is poor (i.e. KGESS < 0) in many catchments in Africa, the North American Great Plains extending into Mexico, with notable patches in eastern Brazil, Thailand, and southern Spain. Results will be biased towards regions with a larger

number of stations, especially when good performing large basins contain many sub-catchments (e.g. Amazon and Rhine basins).

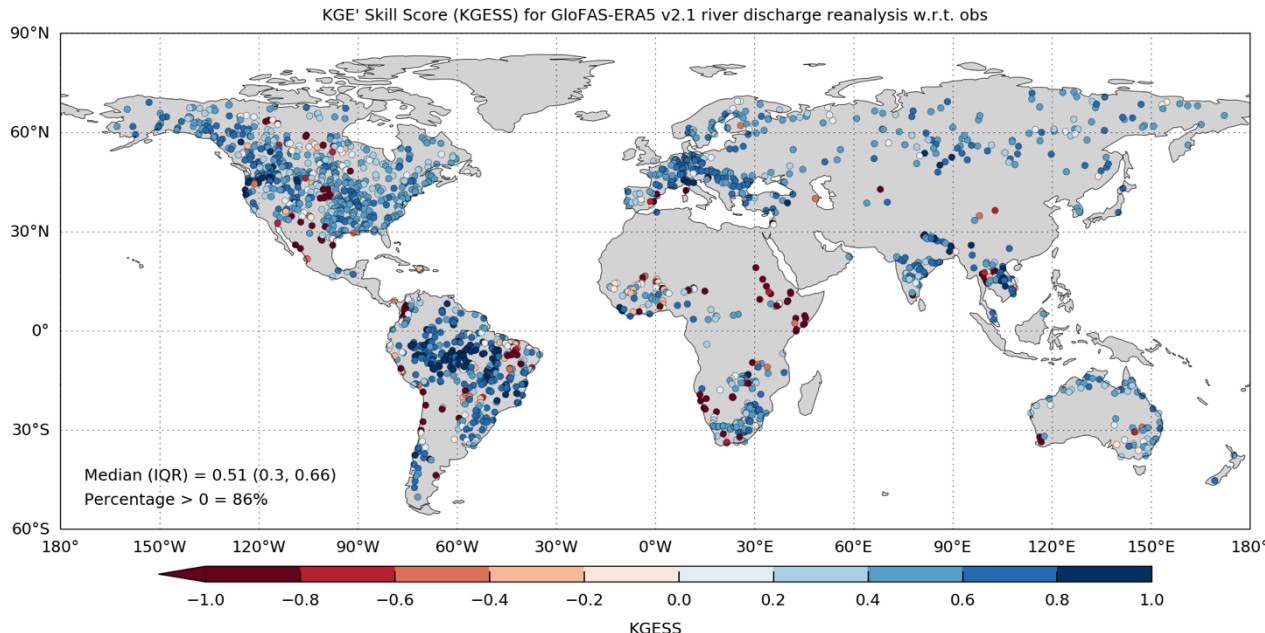

**Figure 4: Modified Kling-Gupta Efficiency Skill Score (KGESS) for GloFAS-ERA5 river discharge reanalysis against 1801 observation stations. Optimum value of KGESS is 1. Blue (red) dots show catchments with positive (negative) skill.**

### 230  4.2 Decomposition into correlation, bias, and variability

An advantage with the KGE' is that it can be decomposed into three constituent components so that greater insights can be gained into which aspects of the GloFAS-ERA5 reanalysis are driving poor and good skill. Almost all (99%) catchments show positive correlation (Figure 5a) with a global median Pearson correlation coefficient of 0.61 (IQR = 0.44, 0.74). Figure 5b shows that river discharge reanalysis is negatively biased in 64 % of catchments (i.e. bias ratio < 1) with global median bias

(expressed as PBIAS) of -16 % (IQR = -38 %, 21 %). In the evaluation of their global river simulation, Lin et al. (2019) consider a PBIAS within ±20 % to be very good. Whilst only 28 % of stations meet this criterion for the GloFAS-ERA5 reanalysis, results are in line with simulations in Lin et al. (2019). Worst performing catchments (dark red KGESS dots in Figure 4) are predominantly driven by very large positive biases (dark blue dots in Figure 5b) in dryer rivers of Central US, Africa, eastern Brazil, as well as the western coast of South America; in total 12 % of catchments have positive PBIAS > 100

240  % (i.e. bias ratio > 2). Figure 5c shows lower variability in GloFAS-ERA5 reanalysis than observations in 61 % of catchments (i.e. variability ratio < 1) but errors in variability are less severe than bias errors with global median values of -9 % (IQR = -31 %, 15 %).

**Figure 5: Decomposition of the Modified Kling-Gupta Efficiency KGE' into its three components, Pearson correlation (a), bias ratio (b), and variability ratio (c) for GloFAS-ERA5 river discharge reanalysis against 1801 observation stations. Optimum values for each of the three KGE' components is 1. Blue (red) dots represent positive (negative) values.**



It is important to also look at the average magnitude of errors as a small over/under estimation in dry rivers can produce large
percentage biases (and hence bias ratios). This was done by converting the units of both the reanalysis and observation time-
series from $m^3 s^{-1}$ to runoff depth across the catchment area in $mm d^{-1}$ to allow direct comparison between catchments of
different sizes, then compute the Mean Absolute Error (MAE) metric (Figure 6). The global median MAE is 0.41 $mm d^{-1}$ (IQR
= 0.18 $mm d^{-1}$, 0.72 $mm d^{-1}$). Most areas with a PBIAS > 100 % (in Fig. 5b), namely much of Africa, central US, and eastern
Brazil, have in fact a low absolute magnitude of errors given their dry locations. Other notable areas with low absolute
magnitude of errors include large parts of India, South East Asia, and Australia. There are however catchments in the western
coast of South America, Sudan and Ethiopia, and tributaries of the River Ganges with a large MAE.

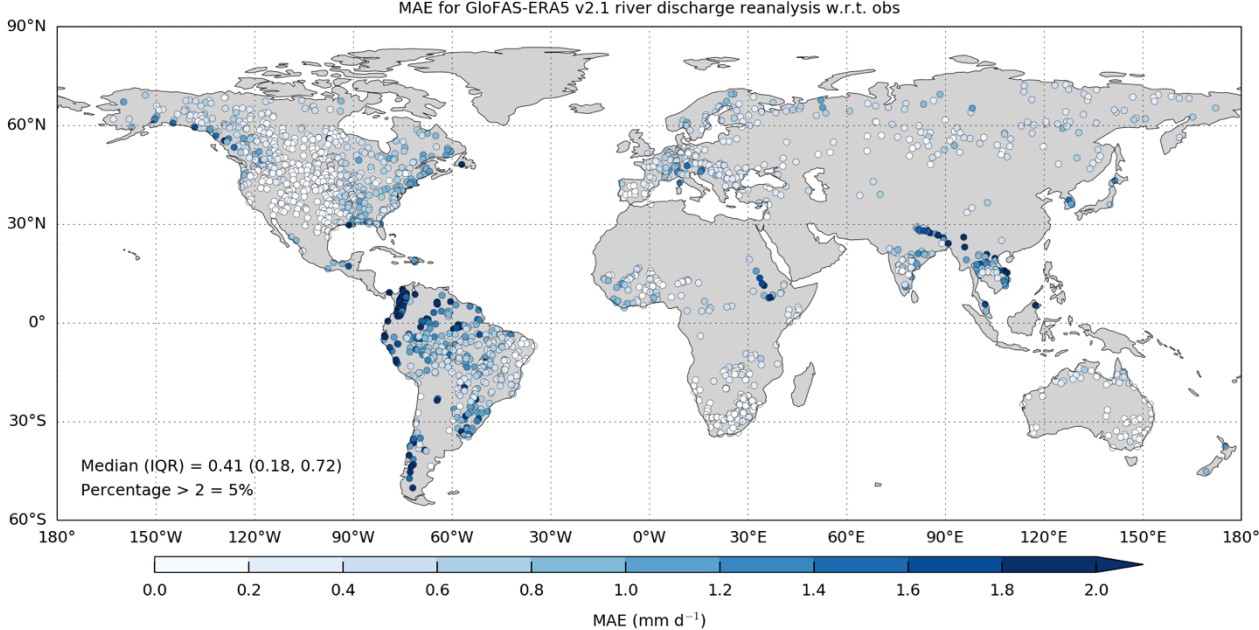

**Figure 6: Mean Absolute Error (MAE) for GloFAS-ERA5 reanalysis against 1801 observation stations. Units for both reanalysis**
**and observations have been converted from $m^3 s^{-1}$ to runoff depth across the catchment area ($mm d^{-1}$) to allow direct comparison**
**of the magnitude of errors. Optimum value of MAE is 0, catchments with larger magnitude of errors are darker shades of blue dots.**



## 4.3 Performance by month

Figure 7 shows the performance of GloFAS-ERA5 reanalysis for each month. Hydrological simulation skill is relatively consistent across each month with median KGESS ranging between 0.32 to 0.41 (Figure 7a). The April to October months have highest skill, with January, February, March, November, and December having a higher proportion of catchments with negative skill. When the KGE' is decomposed into correlation, bias, and variability components at the monthly scale (Figure 7b-d, respectively) it shows that the months with higher incidence of negative KGESS are driven by a higher proportion of

catchments with large positive biases in those months. Correlation and variability error metrics do not vary much from one month to the next, in comparison to bias errors.

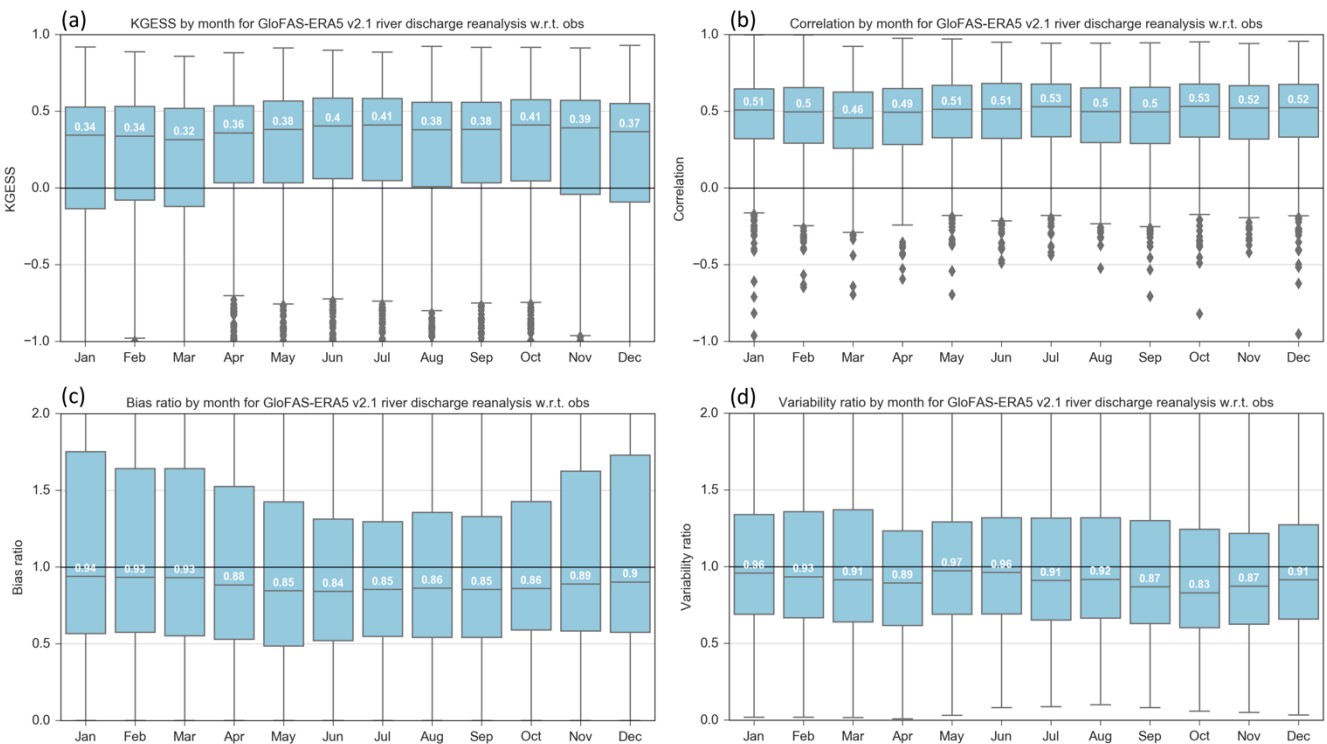

**Figure 7: Performance metrics for each month. Modified Kling-Gupta Efficiency Skill Score (KGESS) (a) with decomposition of KGE' into Pearson correlation (b), bias ratio (c), and variability ratio (d). Boxes represent the IQR and horizontal grey line the**
275 **median. Whiskers extend to the most extreme data point, unless the data point is more than 1.5 times the IQR from the box and is instead represented as an outlier (grey diamond).**

## 4.4 Performance by catchment area

The skill of GloFAS-ERA5 river discharge reanalysis grouped into seven catchment area categories is shown in Fig. 8. In general, skill is lowest for catchments in the three categories < 10,000 $km^2$ with median KGESS = 0.21 (n=39), 0.4 (n=41),

and 0.42 (n=53), respectively. Performance improves as catchment size increases, with median KGESS = 0.56 for catchments > 50,000 $km^2$. It must be noted that results are affected by uneven samples of catchment sizes available within the GloFAS



observations database, with catchments between 10,000 and 50,000 km$^2$ being dominant (n=1013) and smaller catchments being underrepresented.

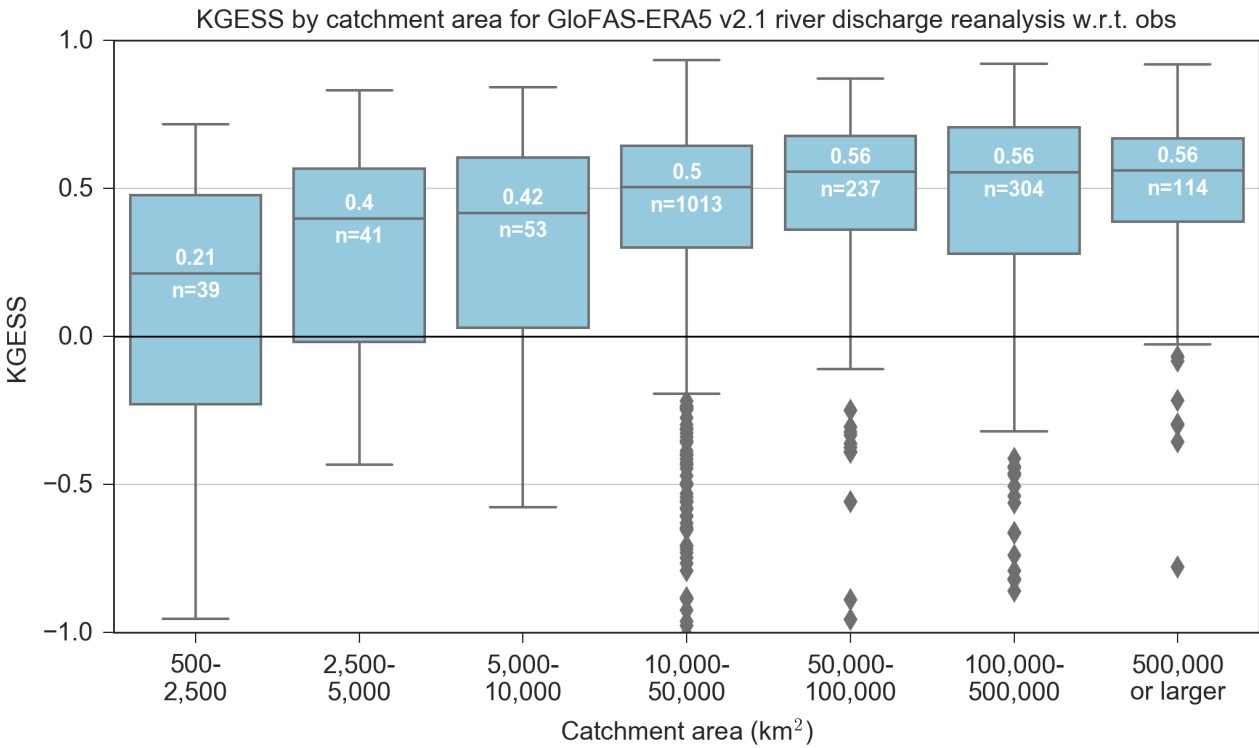

Figure 8: Modified Kling-Gupta Efficiency Skill Score (KGESS) grouped into seven catchment area categories. Boxes and whiskers description as in Fig. 7.

### 4.5 Limitations

This first evaluation has found the dataset to be hydrologically skilful in the vast majority of catchments tested, although the strength of skill can vary considerably depending on location. The degradation in skill, as defined using the KGESS, is the combination of (lower) correlation, (larger) bias errors, and (larger) variability errors. The evaluation provides users with an overview of the global scale quality of the dataset, although users are advised to undertake more in-depth evaluation of the dataset for their region of interest. A key limitation of the dataset is the large biases identified in several regions (see above). Attribution of such biases in the GloFAS-ERA5 reanalysis is outside the scope of this data paper, but ongoing investigations such as Zsoter et al. (2019) has shown biases can be introduced by the real time land data assimilation within the HTESSEL land surface model. Another expected cause of differences between river discharge reanalysis and observations is due to human modification within catchments and river channels (e.g. Harrigan et al., 2014). It is estimated that just 37 % of rivers remain free-flowing globally, with construction of reservoirs and dams the main contributor to loss of connectivity (Grill et al., 2019). While GloFAS-ERA5 reanalysis does represent major dams and reservoirs on the modelled river network, it does so in a



simplified way and does not include operational operating schedules for individual structures. Given the fundamental

dependence of the dataset on ERA5, it would be pertinent for users to be aware of the known ERA5 issues, which can be found in the ERA5 documentation: https://confluence.ecmwf.int/display/CKB/ERA5. In particular, 'rain bombs' are known to occur from time to time in the numerical weather prediction (NWP) model used by ERA5 whereby extremely large rainfall totals are generated, although these are rare and happen over very small areas. However, their impact on hydrology has not been assessed. As with any reanalysis product, care must be taken when calculating long-term trends in river discharge as discontinuities may

be present in the record due to changes in the global observing system entering ERA5.

## 5. Data availability

The GloFAS-ERA5 river discharge reanalysis is provided through the European Commission Copernicus Emergency Management Service (CEMS) and follows the Copernicus open data policy that users shall have free, full, and open access to Copernicus Service Information. With the drive for open data, comes challenges. In the era of 'big data' it is clear that

traditional ways of hosting and disseminating large earth system datasets is no longer fit-for-purpose. An exciting development in the way large climate datasets are discovered, accessed, and used is the Copernicus Climate Change Service (C3S) Climate Data Store (CDS; https://cds.climate.copernicus.eu/cdsapp#!/home). The CDS hosts various global and regional reanalysis products, gridded records for Essential Climate Variables (ECVs), in which river discharge is included as a key terrestrial ECV, and much more. The CDS requires standardisation of data and metadata so that datasets are more useable and

discoverable through the CDS metadata pages. The CDS website provides easy access to data through user-friendly download forms. There is also a CDS Python Application Programming Interface (API) to allow programmatic access to data. An innovative feature of the CDS is the Toolbox, which makes it easier to handle large volumes of data by allowing users to make custom applications, filter data by geographical region and date range, and finally present the data using maps and charts directly through the CDS cloud infrastructure.

The GloFAS-ERA5 river discharge reanalysis product is available on the CDS: https://cds.climate.copernicus.eu/cdsapp#!/dataset/cems-glofas-historical?tab=overview with the following DOI: 10.24381/cds.a4fdd6b9 (C3S, 2019). The CDS landing page for the GloFAS-ERA5 reanalysis dataset is shown in Fig. 9. Both the long-term consolidated and the near real time intermediate reanalysis data are available in two ways. First, through the

325 'Data Download' tab whereby users can manually select options in a form for which data they would like to download. Second, data can be retrieved through the dedicated Python CDS API; an example API retrieval script is shown in Appendix B. Note that users must register for a CDS account (for free) before gaining access.




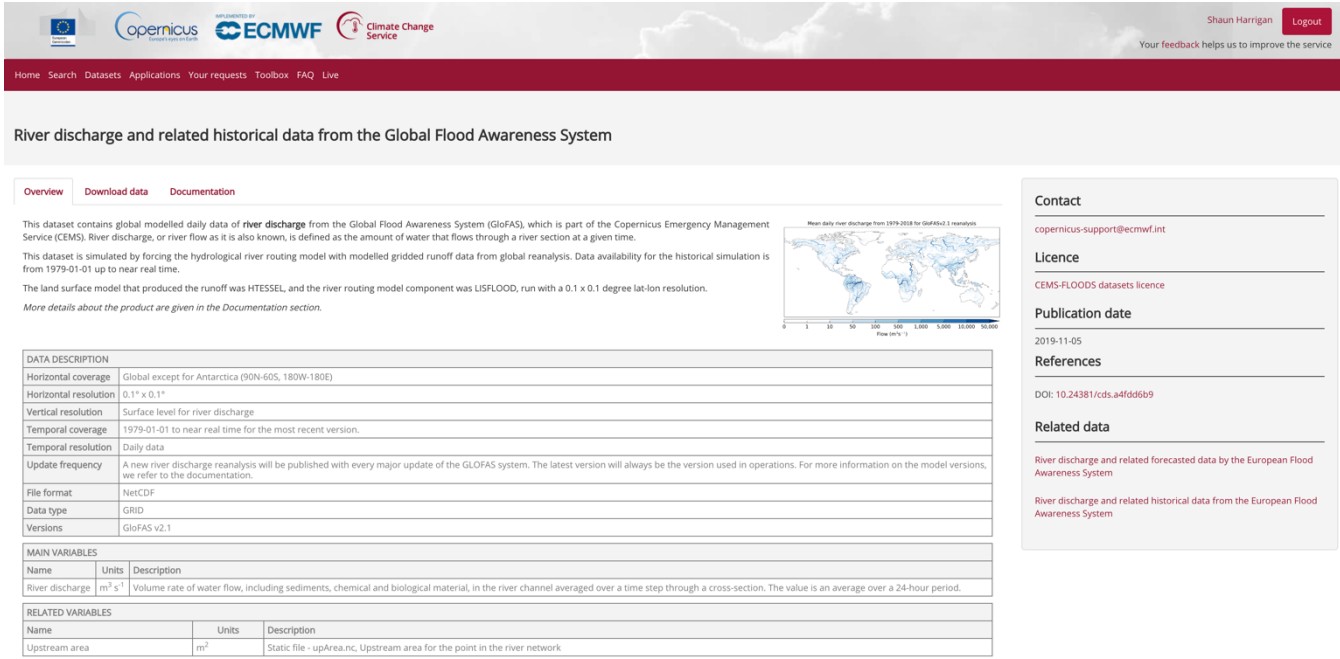

**Figure 9: The GloFAS-ERA5 river discharge reanalysis landing page on the C3S Climate Data Store (CDS:**
**https://cds.climate.copernicus.eu/cdsapp#!/dataset/cems-glofas-historical?tab=overview).**

## 6. Conclusions

This paper outlines the production, description, evaluation, and access to the new GloFAS-ERA5 operational global river discharge reanalysis dataset available from 1979 and updated in near real time. This dataset is central to two key steps within GloFAS, i.) calculation of flood thresholds against which real time ensemble forecasts are compared to determine the probability of a flood signal, and ii.) more consistent hydrometeorological initial conditions for the real time flood and seasonal forecasts. The evaluation against observations showed that the product is skilful in 86 % of catchments according to the modified Kling-Gupta Efficiency Skill Score against a mean flow benchmark. However, skill varies considerably with location with several regions such as central US, Africa, eastern Brazil, and western coast of South America having large systematic positive biases. The results from the evaluation are comparable with other long-term global river discharge products (e.g. Lin et al., 2019). The attribution of such biases in the GloFAS-ERA5 reanalysis is outside the scope of this data paper, but ongoing investigations such as Zsoter et al. (2019) on the biases introduced by the real time land data assimilation within the HTESSEL land surface model will help better understand existing limitations. GloFAS is an operational system which undergoes constant developments with intensive research on future versions of the model. It is foreseen that a new model version will be made operational in 2020 based on the full LISFLOOD hydrological model and an improved model calibration (Alfieri, et al. 2019).



The long-term and operational nature of the GloFAS-ERA5 reanalysis dataset opens avenues for further applications. Forecast evaluation activities within GloFAS now include skill assessment over longer time periods and has allowed a new operational forecast verification suite to be developed whereby the performance of the forecasts can be tracked in near real time for every river in the world. Other applications are envisaged for monitoring the global status of flood and drought conditions, identification hydroclimatic variability and change, and as raw input to post-processing and machine learning methods that can add further value.

## Appendix A

**Standard NetCDF header metadata information for example file**

```
$ ncdump -h CEMS_ECMWF_dis24_20190101_glofas_v2.1.nc

netcdf CEMS_ECMWF_dis24_20190101_glofas_v2.1 {
dimensions:
        time = UNLIMITED ; // (1 currently)
        lon = 3600 ;
        lat = 1500 ;
variables:
        double time(time) ;
                time:standard_name = "time" ;
                time:long_name = "time" ;
                time:units = "hours since 1979-01-01 00:00:00" ;
                time:calendar = "standard" ;
                time:axis = "T" ;
        double lon(lon) ;
                lon:standard_name = "longitude" ;
                lon:long_name = "longitude" ;
                lon:units = "degrees_east" ;
                lon:axis = "X" ;
        double lat(lat) ;
                lat:standard_name = "latitude" ;
                lat:long_name = "latitude" ;
                lat:units = "degrees_north" ;
                lat:axis = "Y" ;
        float dis24(time, lat, lon) ;
                dis24:long_name = "mean discharge in the last 24 hours" ;
                dis24:units = "m3/s" ;
                dis24:_FillValue = 1.e+20f ;
                dis24:missing_value = 1.e+20f ;
```

The 'ncdump' command-line utility converts NetCDF data to human-readable text form

Summary of number of time and space dimensions. This file is for one day

'time' variable with units in hours since reference time. In this e.g. it is 350640 h (or 14,610 d)

Longitude ('lon') and latitude ('lat') variables for grid cells in the GloFAS domain

Primary variable: mean daily river discharge ('dis24') with dimensions ('time', 'lat', 'lon'). Units & missing value also shown



**Appendix B**

```
# Example CDS Python API request script

# Code snippets can be found by clicking 'Show API request' at
# bottom of GloFAS-ERA5 reanalysis download form:
# https://cds.climate.copernicus.eu/cdsapp#!/dataset/cems-glofas-historical?tab=form

# Instructions on how to download CDS API can be found here:
# https://cds.climate.copernicus.eu/api-how-to

import cdsapi

c = cdsapi.Client()

# Example download consolidated data (GloFAS)for 31 December 2018 (note: date stamp
# represents end of 24 h averaging period)

c.retrieve(
    'cems-glofas-historical',
    {
        'variable':'River discharge',
        'dataset':'Consolidated reanalysis',
        'version':'2.1',
        'year':'2019',
        'month':'01',
        'day':'01',
        'format':'tgz'
    },
    'download.tar.gz')

# Example download near real time intermediate data (GloFAST)for 12 November 2019 (note:
# date stamp represent end of 24 h averaging period)

c.retrieve(
    'cems-glofas-historical',
    {
        'variable':'River discharge',
        'dataset':'Intermediate dataset',
        'version':'2.1',
        'year':'2019',
        'month':'11',
        'day':'13',
        'format':'tgz'
    },
    'download.tar.gz')
```

**Author contributions.** SH drafted the manuscript and performed the evaluation. EZ wrote the suite to produce the dataset. CB adapted the suite to produce the dataset operationally. FW and CB were responsible for ingestion of the dataset into the Climate Data Store. LA, CP, PS, HC, and FP helped frame the paper. All co-authors contributed to the editing of the manuscript and to the discussion and interpretation of results.



**Acknowledgements.** We thank colleagues from the Copernicus Climate Change Service (C3S) for helping with ingesting the dataset into the Climate Data Store (CDS). This work was funded by European Commission Copernicus Emergency Management Service (CEMS) Framework Contract No. 198702 awarded to ECMWF. The providers of observed river discharge observations are greatly acknowledged, both GloFAS partners and the Global Runoff Data Centre (GRDC), 56068 Koblenz, Germany.

**Competing interests.** The authors declare that they have no conflict of interest.

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
