# Peer review of "GloFAS-ERA5 operational global river discharge reanalysis 1979present"

_Earth System Science Data, 2019_

## Referee Comment (RC1) · Anonymous Referee #1 · 31 Mar 2020

In this paper, the authors reported a newly developed river discharge dataset at the global scale using a meteorological reanalysis dataset and evaluated its performance. Since this river discharge dataset is very promising in terms of high resolution (0.1 degree) and feasibility for real-time update, it will benefit potential users among hydrology and related-field communities. The contents of this paper are also suited for this journal. This paper is well organized. But there is some room for improvement before publication. In particular, since this paper targets a new release of river discharge data and is intended to be published in the journal specialized for scientific data, methods and processes used in producing the dataset should be solidly and clearly written.

MAJOR COMMENTS

Figure 1: Since the spatial resolution of LISFLOOD (0.1deg) is finer than that of ERA5

[Figure]

runoff data (0.28125deg), I guess a kind of downscaling techniques was used to produce the LISFLOOD dataset. However, there is no information (except "been resampled" in L138) on this process in this paper. How did the authors produce runoff data at a finer resolution in this paper? Did the authors weight the ERA5 runoff value (by something) during the "resampling"? Did the authors consider terrain effects within an ERA5 cell in allocating surface/subsurface runoff to multiple 0.1deg land cells? Please provide the procedure in detail.

Sect. 2.2 and Figure 1: The authors describe surface and subsurface runoff data originally generated from the HTESSEL land model. I think the runoff scheme directly affects the river discharge data, but less information about it is provided. To which depth of soil layer did the authors consider as the subsurface runoff? Regarding the description in L125-127, how much delays were considered before the subsurface water returns back to the river channel in the LISTFLOOD ground water module? Does it depend on the soil properties?

L132-135: The authors describe flow alteration by lakes and reservoirs, but readers cannot figure out how much the flow is altered by them. Did the authors use a kind of algorithms of flow alteration or dam manipulation? The authors also discuss the limitation of this dataset as "While GloFAS-ERA5 reanalysis does represent major dams and reservoirs on the modelled river network, it does so in a simplified way and does not include operational operating schedules for individual structures. (L298-299)" in a later section, but due to the lack of description on dam operation schemes employed in this paper, it is very difficult to have a clear image on that. What does "a simplified way" mean? In addition, how the authors treat river water withdrawal from rivers for human activities (agriculture, industrial, etc.) in this dataset? Please provide information about it in detail.

Sect. 4.3: The authors provide monthly performance of this dataset. Such information is very useful, however, it is very difficult to interpret this seasonality, because the results are (probably) a mixture of contributions from both the northern and southern

hemispheres. Have the authors made similar analysis for each hemisphere? The authors state "Attribution of such biases in the GloFAS-ERA5 reanalysis is outside the scope of this data paper (L293)", however, practical information on the seasonal performance of this dataset will be very beneficial for potential data users. In my view, the authors should add and show, at least, whether a larger bias ratio observed in the months of November to March than the other months (Fig 7c) is attributable to winter discharge from the northern hemisphere or summer discharge from the southern hemisphere (or a mixture of them; or from some specific regions).

MINOR COMMENTS

L139: Is a one-year spin-up enough for this simulation? Probably this depends on the groundwater module or dam operation schemes (the information is not clearly written in the current manuscript, though) used in this model.

L191: The authors used "1801 catchments" here, but this expression might be confusing if there are multiple gauge stations in a large river system. I think dividing this sentence into two parts (and used "1801 stations" in the former one) will be clearer for understanding.

Sect. 4.2: The authors discuss the results by using both the bias ratio (beta) and PBIAS, but this might be confusing. For example, "-9%" in L241 is PBIAS, due to its negative value, I guess.

---

## Referee Comment (RC2) · Anonymous Referee #2 · 9 Apr 2020

The paper entitled "GloFAS-ERA5 operational global river discharge reanalysis 1979-present" presented by Harrigan et al., describes re-analysis driven global river discharge simulations that are updated in near real time and distributed through the Copernicus Climate Change Service Climate Data Store. Overall, the paper is well written and provides the reader with an overview on the methods used for data production, file formats and the performance of the data set.

Given, that this paper is a data-descriptor and neither a model documentation nor a research article there is little to criticize. Nonetheless, some aspects of the paper would benefit from additional information. My main points are summarized below:

1. Terminology: The data product presented is referred to as "reanalysis". Although the runoff data used to drive lisflood stem from a reanalysis, the presented data product

is not an integral part of ERA5. In addition, observational discharge data are only used for calibrating lisflood, but are (to my understanding) not assimilated through a state-updating procedure. Given that the term reanalysis is often associated with state updating, I would find a clarification of the chosen terminology helpful in order to avoid confusion about the nature of the presented data set.

2. Transparency of the data production process: Although the paper does a good job in summarizing the workflow resulting in the presented data set, the amount of information presented is not sufficient to replicate the data. While I acknowledge that a description of ERA5 or Lisflood are beyond the scope of the paper, there are a number of essential technical steps that are not described. Open questions include, but are not limited to, (i) how was ERA5 output disaggregated to the finer resolution, (ii) how was lisflood calibrated (are the data used for validation independent of the data used for calibration), (iii) what does it mean that reservoirs are included (e.g. is management also simulated), etc. I realize that some of these questions are also treated in other publications but for a user of the data set a comprehensive overview with more details would be essential to fully understand the capabilities (and limitations) of the data.

3. The output variable (discharge) is "Volume rate of water flow, including sediments, . . .". While I acknowledge that this is likely the variable of interest for flood forecasting, I would appreciate if the volume (or mass) of pure $H_2O$ could also be made available (if this does not differ significantly, then a statement explaining this might be useful).

4. What is the time resolution of the observations used for validation? I assume daily, but this was not stated explicitly.

5. Stations used for evaluation come "predominantly" from the GRDC. This is not transparent at all and hinders reproducibility of the study. I assume that some of the data cannot be re-distributed, but an overview (e.g. supplementary table) on the considered stations including some key properties (geolocation, river and station names, data-provider, catchment area, . . .) foster reproducibility of the results.

6. If there is more than one station per grid-cell only one station is selected. This is OK. However, what is the criterion to select a particular station (random, expert judgment, catchment size,. . .)?

7. I personally would find extended global summaries (in addition to medians and IQRs) of the performance metrics useful (e.g. tables with percentiles, or empirical cumulative distribution functions).

8. The performance assessment focusses predominantly on the skill of the full time series at daily resolution. For some users information focussing on different modes of variability (e.g. seasonal cycle, anomalies of the seasonal cycle, year-to-year fluctuations) would be also of great interest.

9. Accessibility of the data product. I am aware of and support the effort of the Copernicus Climate Change Service but I don't have an account for this at the time being. I am also reluctant to create "random" accounts that I need to keep track of if not really needed. Given the fact that the data are produced by one of the world leading institutions for global weather data (ECMWF) and hare hosted on the Copernicus platform, I assume that the data format will be state of the art.

---

## Author Comment (AC1) · 24 Jun 2020

**Response to Anonymous Referee #1**

In this paper, the authors reported a newly developed river discharge dataset at the global scale using a meteorological reanalysis dataset and evaluated its performance. Since this river discharge dataset is very promising in terms of high resolution (0.1 degree) and feasibility for real-time update, it will benefit potential users among hydrology and related-field communities. The contents of this paper are also suited for this journal. This paper is well organized. But there is some room for improvement before publication. In particular, since this paper targets a new release of river discharge data and is intended to be published in the journal specialized for scientific data, methods and processes used in producing the dataset should be solidly and clearly written.

Thank you for your positive comments and constructive feedback. We believe your clarifications have sharpened the manuscript and made it clearer for the reader. Our responses to your comments are provided below in blue together with your original review in black.

MAJOR COMMENTS

Figure 1: Since the spatial resolution of LISFLOOD (0.1deg) is finer than that of ERA5 (0.28125deg), I guess a kind of downscaling techniques was used to produce the LISFLOOD dataset. However, there is no information (except "been resampled" in L138) on this process in this paper. How did the authors produce runoff data at a finer resolution in this paper? Did the authors weight the ERA5 runoff value (by something) during the "resampling"? Did the authors consider terrain effects within an ERA5 cell in allocating surface/subsurface runoff to multiple 0.1deg land cells? Please provide the procedure in detail.

This question was also asked by Anonymous Referee #2. In order to be consistent with the operational GloFAS procedure, the runoff fields from ERA5 were downscaled using the simple nearest neighbour method from the native resolution to the 0.1° LISFLOOD grid. The task was done using the open source 'pyg2p' module with interpolation 'grib_nearest' option in Python (https://pypi.org/project/pyg2p/). No weights were applied to runoff values and terrain effects within the ERA5 cell were not considered. We will add the additional detail on how the downscaling was done in the resubmitted manuscript.

Sect. 2.2 and Figure 1: The authors describe surface and subsurface runoff data originally generated from the HTESSEL land model. I think the runoff scheme directly affects the river discharge data, but less information about it is provided. To which depth of soil layer did the authors consider as the subsurface runoff? Regarding the description in L125-127, how much delays were considered before the subsurface water returns back to the river channel in the LISTFLOOD ground water module? Does it depend on the soil properties?

The HTESSEL land surface model is used to calculate the water balance at the land surface. Excess precipitation and snowmelt are partitioned as surface runoff or infiltrated into a four-layer soil column (7 cm depth for top layer and then 21, 72, and 189 cm) at each ERA5 grid cell, before draining from the bottom of the soil column as sub-surface runoff. Water moving (vertically) through the soil column does depend on soil properties. There are six soil texture classes in HTESSEL (e.g. coarse or very fine) that determine hydraulic properties. Therefore, the soil properties will determine the amount of time it takes for water to exit at the bottom of the soil column as sub-surface runoff. Further information on the details is provided in Balsamo et al. (2009).

As mentioned in L125-127, the HTESSEL sub-surface runoff is used as input to the LISFLOOD groundwater module, which consists of two parallel linear reservoirs (upper zone for quick and lower zone for slower groundwater flow) that store and subsequently transport water to the river channel with a time delay. In Hirpa et al. (2018), the upper zone time constant was given a default value of 10 days with a lower (upper) bound of 3 days (40 days) during calibration. The upper zone time constant has a default value of 200 days with a lower (upper) bound of 40 days (500 days) during calibration.

L132-135: The authors describe flow alteration by lakes and reservoirs, but readers cannot figure out how much the flow is altered by them. Did the authors use a kind of algorithms of flow alteration or dam manipulation? The authors also discuss the limitation of this dataset as "While GloFAS-ERA5 reanalysis does represent major dams and reservoirs on the modelled river network, it does so in a simplified way and does not include operational operating schedules for individual structures. (L298-299)" in a later section, but due to the lack of description on dam operation schemes employed in this paper, it is very difficult to have a clear image on that. What does "a simplified way" mean? In addition, how the authors treat river water withdrawal from rivers for human activities (agriculture, industrial, etc.) in this dataset? Please provide information about it in detail.

Reservoir outflow is calculated with a set of simplified rules depending on their filling level, and balances water recharge if storage is below normal or release if above normal. There is a minimum outflow to ensure the downstream river does not dry up, and a non-damaging release so the reservoir does not reach full capacity. Simplified reservoir operating parameters were used based on expert opinion (outlined in Zajac et al., 2017) given lack of availability of global operational release records.

As mentioned in our reply to Anonymous Reviewer #2, we propose to add a new table (Table A below) to accompany Fig. 1 in the resubmitted manuscript that makes it clearer for the reader to find the open access publications outlining the full methodological details of the key components of GloFAS-ERA5:

**Table A: Scientific papers and model documentation for the key components in the production of GloFAS-ERA5 v2.1 river discharge reanalysis dataset.**

| GloFAS-ERA5 component | Description | Reference |
|---|---|---|
| ERA5 | Global reanalysis dataset using ECMWF Integrated Forecast System (IFS) model cycle 41r2 from 1979 to present | Hersbach et al. (2020) |
| ERA5 runoff | Surface and sub-surface runoff within ERA5 generated using the HTESSEL land surface model | Balsamo et al. (2009) |
| LISFLOOD river discharge | River discharge generated using LISFLOOD hydrological and channel routing model to route runoff into and through the river network and provide groundwater storage. LISFLOOD includes lake, reservoir and human water use routines | Burek et al. (2013) |
| Lakes and reservoirs used | Incorporated 463 lakes and 667 reservoirs | Zajac et al. (2017) |

| | | |
|---|---|---|
| in GloFAS | into the GloFAS river network | |
| Calibration of LISFLOOD used in GloFAS | LISFLOOD was calibrated against daily river discharge from 1287 observation stations worldwide | Hirpa et al. (2018) |

Sect. 4.3: The authors provide monthly performance of this dataset. Such information is very useful, however, it is very difficult to interpret this seasonality, because the results are (probably) a mixture of contributions from both the northern and southern hemispheres. Have the authors made similar analysis for each hemisphere? The authors state "Attribution of such biases in the GloFAS-ERA5 reanalysis is outside the scope of this data paper (L293)", however, practical information on the seasonal performance of this dataset will be very beneficial for potential data users. In my view, the authors should add and show, at least, whether a larger bias ratio observed in the months of November to March than the other months (Fig 7c) is attributable to winter discharge from the northern hemisphere or summer discharge from the southern hemisphere (or a mixture of them; or from some specific regions).

This is a very good suggestion. We have conducted your proposed analysis (Fig. A) and found that while the overall GloFAS-ERA5 monthly performance in each hemisphere does not change substantially from the global analysis (Fig. 7 in the paper), there are some differences worth reporting. We therefore propose to add Fig. A to the resubmitted manuscript as a new Fig. 8 together with the following additional paragraph in Sect. 4.3:

"Results are grouped into northern (n=1268 stations) and southern (n=533 stations) hemispheres in Fig. 8. The overall GloFAS-ERA5 monthly performance in each hemisphere does not change substantially from the global analysis (Fig. 7). Nevertheless, there are some differences. The KGESS and bias ratio from the northern hemisphere (Fig. 8a and c, respectively) tend to follow the global analysis most strongly (i.e. Fig. 7a and c, respectively), which is not surprising given 70 % of all stations are located in the northern hemisphere. However, a higher proportion of southern hemisphere stations show large positive biases from April to June, compared to November to March in the northern hemisphere. The largest proportion of stations with negative KGESS in the southern hemisphere are found from August to October (Fig. 8a). These months correspond with lower southern hemisphere correlation (Fig. 8b) and a higher proportion of stations with large positive variability ratios (i.e. GloFAS-ERA5 has higher variability than observed river discharge)."

[Figure]

**Figure A: Performance metrics for each month by hemisphere. Modified Kling-Gupta Efficiency Skill Score (KGESS) (a) with decomposition of KGE' into Pearson correlation (b), bias ratio (c), and variability ratio (d). Brown (green) boxes represent the IQR and horizontal grey line the median for the northern (southern) hemisphere. Whiskers extend to the most extreme data point, unless the data point is more than 1.5 times the IQR from the box and is instead represented as an outlier (grey diamond).**

MINOR COMMENTS L139:

Is a one-year spin-up enough for this simulation? Probably this depends on the groundwater module or dam operation schemes (the information is not clearly written in the current manuscript, though) used in this model.

The initialisation routine within LISFLOOD has been designed so that one year is sufficient to estimate the initial state of all the state variables. For state variables that are fast responding and can reach equilibrium quickly (e.g. storage in the upper groundwater zone), a value of 0 can be used at start of the run and one year is more than long enough. However, for other state variables that are slowly responding it is correct that one year can be too short in some model calibration routines, especially for catchments with large groundwater stores. In order to avoid the need for very long spin up periods, LISFLOOD calculates a "steady-state" storage amount for the lower groundwater zone during a long-term pre-run, and thus reduces the lower zone's spin up time (Burek et al., 2013). We will add a sentence on this to the resubmitted manuscript.

L191: The authors used "1801 catchments" here, but this expression might be confusing if there are multiple gauge stations in a large river system. I think dividing this sentence into two parts (and used "1801 stations" in the former one) will be clearer for understanding.

We will change this to "1801 stations" in the resubmitted manuscript.

Sect. 4.2: The authors discuss the results by using both the bias ratio (beta) and PBIAS, but this might be confusing. For example, "-9%" in L241 is PBIAS, due to its negative value, I guess.

The bias term in the decomposition of the KGE' is the bias ratio $\beta$ but can easily be converted to the more widely used percent bias (PBIAS) by $(\beta - 1) \times 100$ (as shown in L205-206 in the submitted manuscript). PBIAS was also used in Lin et al. (2019) when considering what we mean by a "very good" bias error in global hydrological modelling (i.e. ±20 %), mentioned on L235-236. Our original intention was to report the bias in the text in the more widely used PBIAS form. However, your point is valid that it might actually be more confusing to the reader. We will therefore report bias errors as bias ratios and variability errors as variability ratios as per Equations 2 and 3 in the resubmitted manuscript.

We really appreciate your time and insight in reviewing our manuscript,

Kind regards,
Shaun (on behalf of all co-authors)

**References**

Balsamo, G., Beljaars, A., Scipal, K., Viterbo, P., van den Hurk, B., Hirschi, M. and Betts, A. K.: A Revised Hydrology for the ECMWF Model: Verification from Field Site to Terrestrial Water Storage and Impact in the Integrated Forecast System, J. Hydrometeor., 10(3), 623–643, doi:10.1175/2008JHM1068.1, 2009.

Burek, P., van der Knijff, J. M. and de Roo, A. P. J. D.: LISFLOOD - Distributed Water Balance and Flood Simulation Model - Revised User Manual, Publications Office of the European Union, doi: 10.2788/24719, 2013.

Hersbach, H., Bell, B., Berrisford, P., Hirahara, S., Horányi, A., Muñoz-Sabater, J., Nicolas, J., Peubey, C., Radu, R., Schepers, D., Simmons, A., Soci, C., Abdalla, S., Abellan, X., Balsamo, G., Bechtold, P., Biavati, G., Bidlot, J., Bonavita, M., Chiara, G. D., Dahlgren, P., Dee, D., Diamantakis, M., Dragani, R., Flemming, J., Forbes, R., Fuentes, M., Geer, A., Haimberger, L., Healy, S., Hogan, R. J., Hólm, E., Janisková, M., Keeley, S., Laloyaux, P., Lopez, P., Lupu, C., Radnoti, G., Rosnay, P. de, Rozum, I., Vamborg, F., Villaume, S. and Thépaut, J.-N.: The ERA5 Global Reanalysis, Quarterly Journal of the Royal Meteorological Society, doi:10.1002/qj.3803, 2020.

Hirpa, F. A., Salamon, P., Beck, H. E., Lorini, V., Alfieri, L., Zsoter, E. and Dadson, S. J.: Calibration of the Global Flood Awareness System (GloFAS) using daily streamflow data, J. Hydrol., 566, 595–606, doi:10.1016/j.jhydrol.2018.09.052, 2018.

Zajac, Z., Revilla-Romero, B., Salamon, P., Burek, P., Hirpa, F. A. and Beck, H.: The impact of lake and reservoir parameterization on global streamflow simulation, J. Hydrol., 548, 552–568, doi:10.1016/j.jhydrol.2017.03.022, 2017.

---

## Author Comment (AC2) · 24 Jun 2020

**Response to Anonymous Referee #2**

The paper entitled "GloFAS-ERA5 operational global river discharge reanalysis 1979- present" presented by Harrigan et al., describes re-analysis driven global river discharge simulations that are updated in near real time and distributed through the Copernicus Climate Change Service Climate Data Store. Overall, the paper is well written and provides the reader with an overview on the methods used for data production, file formats and the performance of the data set.

Thank you for your positive comments and constructive feedback. Your clarifications have improved the manuscript and made it clearer for the reader. Our responses to your comments are provided below in blue together with your original review in black.

Given, that this paper is a data-descriptor and neither a model documentation nor a research article there is little to criticize. Nonetheless, some aspects of the paper would benefit from additional information. My main points are summarized below:

1. Terminology: The data product presented is referred to as "reanalysis". Although the runoff data used to drive lisflood stem from a reanalysis, the presented data product is not an integral part of ERA5. In addition, observational discharge data are only used for calibrating lisflood, but are (to my understanding) not assimilated through a state-updating procedure. Given that the term reanalysis is often associated with state updating, I would find a clarification of the chosen terminology helpful in order to avoid confusion about the nature of the presented data set.

We use the term reanalysis to mean the optimal combining of in situ and satellite earth system observations together with models to provide consistent spatio-temporal "maps without gaps" of land, ocean and atmospheric variables of interest as per Hersbach et al. (2020) and is now common in Earth System Modelling.

2. Transparency of the data production process: Although the paper does a good job in summarizing the workflow resulting in the presented data set, the amount of information presented is not sufficient to replicate the data. While I acknowledge that a description of ERA5 or Lisflood are beyond the scope of the paper, there are a number of essential technical steps that are not described. Open questions include, but are not limited to, (i) how was ERA5 output disaggregated to the finer resolution, (ii) how was lisflood calibrated (are the data used for validation independent of the data used for calibration), (iii) what does it mean that reservoirs are included (e.g. is management also simulated), etc. I realize that some of these questions are also treated in other publications but for a user of the data set a comprehensive overview with more details would be essential to fully understand the capabilities (and limitations) of the data.

Your overall point on the need for additional clarity, especially in regards to the hydrological modelling detail, was also raised Anonymous Referee #1. It is indeed a balance within this data paper to focus on the description of the GloFAS-ERA5 dataset and its evaluation, while providing sufficient detail on the modelling methodology to allow users to gain an understanding of the capabilities and limitations. Our intention is to provide only a summary of the modelling methodology that is already described in full detail in the published literature. Given your queries, we propose to include the new Table A (below) to accompany Fig. 1 in the resubmitted manuscript that summarises the published scientific papers and model documentation for each of the key GloFAS-ERA5 components. All these publications are open access.

Responses to your individual queries are given below:

i.) This question was also asked by Anonymous Referee #1. In order to be consistent with the operational GloFAS procedure, the runoff fields from ERA5 were downscaled using the simple nearest neighbour method from the native resolution to the 0.1° LISFLOOD grid. The task was done using the open source 'pyg2p' module with interpolation 'grib_nearest' option in Python (https://pypi.org/project/pyg2p/). No weights were applied to runoff values and terrain effects within the ERA5 cell were not considered. We will add the additional detail on how the downscaling was done in the resubmitted manuscript.

ii.) The LISFLOOD version used here for GloFAS-ERA5 v2.1 was calibrated by Hirpa et al. (2018) using an evolutionary optimisation algorithm against daily river discharge from 1287 stations worldwide. For each station, the record was split in two for calibration and validation. If the record was shorter than eight years, four years were used for calibration and the remainder for validation. If the record was equal to or longer than eight years, half was used for calibration and half for validation, with the most recent period used for calibration.

iii.) Reservoir outflow is calculated with a set of simplified rules depending on their filling level, and balances water recharge if storage is below normal or release if above normal. There is a minimum outflow to ensure the downstream river does not dry up, and a non-damaging release so the reservoir does not reach full capacity. Simplified reservoir operating parameters were used based on expert opinion (outlined in Zajac et al., 2017) given lack of availability of global operational release records.

**Table A: Scientific papers and model documentation for the key components in the production of GloFAS-ERA5 v2.1 river discharge reanalysis dataset.**

| GloFAS-ERA5 component | Description | Reference |
|---|---|---|
| ERA5 | Global reanalysis dataset using ECMWF Integrated Forecast System (IFS) model cycle 41r2 from 1979 to present | Hersbach et al. (2020) |
| ERA5 runoff | Surface and sub-surface runoff within ERA5 generated using the HTESSEL land surface model | Balsamo et al. (2009) |
| LISFLOOD river discharge | River discharge generated using LISFLOOD hydrological and channel routing model to route runoff into and through the river network and provide groundwater storage. LISFLOOD includes lake, reservoir and human water use routines | Burek et al. (2013) |
| Lakes and reservoirs used in GloFAS | Incorporated 463 lakes and 667 reservoirs into the GloFAS river network | Zajac et al. (2017) |
| Calibration of LISFLOOD used in GloFAS | LISFLOOD was calibrated against daily river discharge from 1287 observation stations worldwide | Hirpa et al. (2018) |

3. The output variable (discharge) is "Volume rate of water flow, including sediments, . . .". While I acknowledge that this is likely the variable of interest for flood forecasting, I would appreciate if the volume (or mass) of pure H2O could also be made available (if this does not differ significantly, then a statement explaining this might be useful).

The definition given in Table 2 is the generic definition of discharge from rivers and streams by the World Meteorological Organisation (WMO) for hydrological products (https://www.wmo.int/pages/prog/www/WMOCodes/WMO306_vI2/LatestVERSION/WMO306_vI2_GRIB2_CodeFlag_en.pdf). It is used across all river discharge products at ECMWF and on the Copernicus Climate Change Service (C3S) Climate Data Store (https://apps.ecmwf.int/codes/grib/param-db?id=240013). Virtually all hydrological models, including GloFAS-ERA5, simulate the volume rate of water only due to inherent simplifications of reality.

4. What is the time resolution of the observations used for validation? I assume daily, but this was not stated explicitly.

Yes, the observations are daily, and the evaluation carried out at the daily scale. Both will be clarified in the resubmitted manuscript.

5. Stations used for evaluation come "predominantly" from the GRDC. This is not transparent at all and hinders reproducibility of the study. I assume that some of the data cannot be re-distributed, but an overview (e.g. supplementary table) on the considered stations including some key properties (geolocation, river and station names, data-provider, catchment area, . . .) foster reproducibility of the results.

Observations cannot be redistributed by the authors due to licencing agreements but to foster reproducibility of the results we will include a Supplementary Table S1 in the resubmitted manuscript including the following metadata for each of the 1801 stations: *GloFAS_ID, Provider, Provider_ID, Station_Name, River_Name, River_Basin, Country_Name, Catchment_Area_Provider_km2, Catchment_Area_GloFAS_km2, Latitude_Provider, Longitude_Provider, Latitude_GloFAS, Longitude_GloFAS.*

In addition, we will include in the Supplementary Table the corresponding performance metrics for each station to allow users to explore the results in more detail: *KGE', KGESS, correlation, bias_ratio, variability_ratio, MAE_mm_per_day.*

6. If there is more than one station per grid-cell only one station is selected. This is OK. However, what is the criterion to select a particular station (random, expert judgment, catchment size,. . .)?

When multiple observation stations were matched to the same GloFAS river cell, the station with the longest record was retained. This criterion removed 27 stations from the initial list. This will be qualified in the resubmitted text.

7. I personally would find extended global summaries (in addition to medians and IQRs) of the performance metrics useful (e.g. tables with percentiles, or empirical cumulative distribution functions).

We have taken your suggestion on board and will also present the performance metrics for all 1801 stations as a cumulative distribution function in the resubmitted manuscript (Fig. A). We will also include the performance metrics for each station along with the metadata in a Supplementary Table S1 you suggested in your point number 5.

[Figure]

**Figure A. Cumulative distribution function of performance metrics across all 1801 stations. Modified Kling-Gupta Efficiency (KGE') and Skill Score (KGESS) (a) with decomposition of KGE' into Pearson correlation (b), bias ratio (c), and variability ratio (d). The red dot marks the optimum value for each metric.**

8. The performance assessment focuses predominantly on the skill of the full time series at daily resolution. For some users information focussing on different modes of variability (e.g. seasonal cycle, anomalies of the seasonal cycle, year-to-year fluctuations) would be also of great interest.

Thank you for your suggestion. We agree that there are many other exciting potential applications of GloFAS-ERA5 that would be interested in an aggregation of the dataset. But here, focusing on the daily time-step will provide the performance of the dataset at the highest temporal resolution that is of most interest for the vast majority of hydrological applications. We expect and encourage users to undertake their own local evaluation for their specific application as the use of the dataset grows.

9. Accessibility of the data product. I am aware of and support the effort of the Copernicus Climate Change Service but I don't have an account for this at the time being. I am also reluctant to create "random" accounts that I need to keep track of if not really needed. Given the fact that the data are produced by one of the world leading institutions for global weather data (ECMWF) and hare hosted on the Copernicus platform, I assume that the data format will be state of the art.

Thank you for your positive comment. A requirement for the GloFAS-ERA5 data to be hosted by the C3S Climate Data Store (CDS) is that state-of-the-art cataloguing, data format (i.e. NetCDF), and standardised metadata and documentation are adhered to: https://cds.climate.copernicus.eu/cdsapp#!/dataset/cems-glofas-historical?tab=overview. This allows GloFAS-ERA5 data to be found through the CDS search catalogue, work with the CDS "Toolbox", and follows the protocol that provides programmatic access to the data via the CDS Application Programming Interface (API). These are valuable tools to allow users to work more easily with large global datasets. More general information on the CDS and how data are delivered can be found here: https://climate.copernicus.eu/climate-data-store.

We really appreciate your time and insight in reviewing our manuscript,

Kind regards,
Shaun (on behalf of all co-authors)

**References**

Balsamo, G., Beljaars, A., Scipal, K., Viterbo, P., van den Hurk, B., Hirschi, M. and Betts, A. K.: A Revised Hydrology for the ECMWF Model: Verification from Field Site to Terrestrial Water Storage and Impact in the Integrated Forecast System, J. Hydrometeor., 10(3), 623–643, doi:10.1175/2008JHM1068.1, 2009.

Burek, P., van der Knijff, J. M. and de Roo, A. P. J. D.: LISFLOOD - Distributed Water Balance and Flood Simulation Model - Revised User Manual, Publications Office of the European Union, doi: 10.2788/24719, 2013.

Hersbach, H., Bell, B., Berrisford, P., Hirahara, S., Horányi, A., Muñoz-Sabater, J., Nicolas, J., Peubey, C., Radu, R., Schepers, D., Simmons, A., Soci, C., Abdalla, S., Abellan, X., Balsamo, G., Bechtold, P., Biavati, G., Bidlot, J., Bonavita, M., Chiara, G. D., Dahlgren, P., Dee, D., Diamantakis, M., Dragani, R., Flemming, J., Forbes, R., Fuentes, M., Geer, A., Haimberger, L., Healy, S., Hogan, R. J., Hólm, E., Janisková, M., Keeley, S., Laloyaux, P., Lopez, P., Lupu, C., Radnoti, G., Rosnay, P. de, Rozum, I., Vamborg, F., Villaume, S. and Thépaut, J.-N.: The ERA5 Global Reanalysis, Quarterly Journal of the Royal Meteorological Society, doi:10.1002/qj.3803, 2020.

Hirpa, F. A., Salamon, P., Beck, H. E., Lorini, V., Alfieri, L., Zsoter, E. and Dadson, S. J.: Calibration of the Global Flood Awareness System (GloFAS) using daily streamflow data, J. Hydrol., 566, 595–606, doi:10.1016/j.jhydrol.2018.09.052, 2018.

Zajac, Z., Revilla-Romero, B., Salamon, P., Burek, P., Hirpa, F. A. and Beck, H.: The impact of lake and reservoir parameterization on global streamflow simulation, J. Hydrol., 548, 552–568, doi:10.1016/j.jhydrol.2017.03.022, 2017.

---

## Author Response (AR1)

**Point-by-point response to Harrigan et al. GloFAS-ERA5**

Dear David,

We thank again both Anonymous Referee's for their comments and constructive feedback. We believe their clarifications have sharpened the manuscript and made it clearer for the reader. Our point by point responses to each of their comments are provided below in blue together with their original review in black. Where the text is changed in the resubmitted manuscript is highlighted and a marked up version of the resubmitted manuscript is attached at the end of this document.

**Response to Anonymous Referee #1**

In this paper, the authors reported a newly developed river discharge dataset at the global scale using a meteorological reanalysis dataset and evaluated its performance. Since this river discharge dataset is very promising in terms of high resolution (0.1 degree) and feasibility for real-time update, it will benefit potential users among hydrology and related-field communities. The contents of this paper are also suited for this journal. This paper is well organized. But there is some room for improvement before publication. In particular, since this paper targets a new release of river discharge data and is intended to be published in the journal specialized for scientific data, methods and processes used in producing the dataset should be solidly and clearly written.

Thank you for your positive comments and constructive feedback. We believe your clarifications have sharpened the manuscript and made it clearer for the reader.

MAJOR COMMENTS
Figure 1: Since the spatial resolution of LISFLOOD (0.1deg) is finer than that of ERA5 (0.28125deg), I guess a kind of downscaling techniques was used to produce the LISFLOOD dataset. However, there is no information (except "been resampled" in L138) on this process in this paper. How did the authors produce runoff data at a finer resolution in this paper? Did the authors weight the ERA5 runoff value (by something) during the "resampling"? Did the authors consider terrain effects within an ERA5 cell in allocating surface/subsurface runoff to multiple 0.1deg land cells? Please provide the procedure in detail.

**Response:** This question was also asked by Anonymous Referee #2. In order to be consistent with the operational GloFAS procedure, the runoff fields from ERA5 were downscaled using the simple nearest neighbour method from the native ERA5 to the 0.1° GloFAS grid. The task was done using the open source 'pyg2p' module with interpolation 'grib_nearest' option in Python (https://pypi.org/project/pyg2p/). No weights were applied to runoff values and terrain effects within the ERA5 cell were not considered.

**Change:** We have added the following sentence into L151-153 in the resubmitted manuscript: "In order to be consistent with the operational GloFAS procedure, the runoff fields from ERA5 were downscaled using the simple nearest neighbour method from the native ERA5 to the 0.1° GloFAS grid."

Sect. 2.2 and Figure 1: The authors describe surface and subsurface runoff data originally generated from the HTESSEL land model. I think the runoff scheme directly affects the river discharge data, but less information

about it is provided. To which depth of soil layer did the authors consider as the subsurface runoff? Regarding the description in L125-127, how much delays were considered before the subsurface water returns back to the river channel in the LISTFLOOD ground water module? Does it depend on the soil properties?

**Response:** The HTESSEL land surface model is used to calculate the water balance at the land surface. Excess precipitation and snowmelt are partitioned as surface runoff or infiltrated into a four-layer soil column (7 cm depth for top layer and then 21, 72, and 189 cm) at each ERA5 grid cell, before draining from the bottom of the soil column as sub-surface runoff. Water moving (vertically) through the soil column does depend on soil properties. There are six soil texture classes in HTESSEL (e.g. coarse or very fine) that determine hydraulic properties. Therefore, the soil properties will determine the amount of time it takes for water to exit at the bottom of the soil column as sub-surface runoff. Further information on the details is provided in Balsamo et al. (2009).

As mentioned in L125-127, the HTESSEL sub-surface runoff is used as input to the LISFLOOD groundwater module, which consists of two parallel linear reservoirs (upper zone for quick and lower zone for slower groundwater flow) that store and subsequently transport water to the river channel with a time delay. In Hirpa et al. (2018), the upper zone time constant was given a default value of 10 days with a lower (upper) bound of 3 days (40 days) during calibration. The upper zone time constant has a default value of 200 days with a lower (upper) bound of 40 days (500 days) during calibration.

**Change:** We have added the following sentence into L119-122 in the resubmitted manuscript: "Excess precipitation and snowmelt are partitioned as surface runoff or infiltrated into a four-layer soil column (7 cm depth for top layer and then 21, 72, and 189 cm) at each ERA5 grid cell, before draining from the bottom of the soil column as sub-surface runoff (Balsamo et al., 2009)".

We have added the following sentence into L139-141 in the resubmitted manuscript: "In Hirpa et al. (2018), the upper zone time constant was given a default value of 10 days with a lower (upper) bound of 3 days (40 days) during calibration. The upper zone time constant has a default value of 200 days with a lower (upper) bound of 40 days (500 days) during calibration".

L132-135: The authors describe flow alteration by lakes and reservoirs, but readers cannot figure out how much the flow is altered by them. Did the authors use a kind of algorithms of flow alteration or dam manipulation? The authors also discuss the limitation of this dataset as "While GloFAS-ERA5 reanalysis does represent major dams and reservoirs on the modelled river network, it does so in a simplified way and does not include operational operating schedules for individual structures. (L298-299)" in a later section, but due to the lack of description on dam operation schemes employed in this paper, it is very difficult to have a clear image on that. What does "a simplified way" mean? In addition, how the authors treat river water withdrawal from rivers for human activities (agriculture, industrial, etc.) in this dataset? Please provide information about it in detail.

**Response:** Reservoir outflow is calculated with a set of simplified rules depending on their filling level, and balances water recharge if storage is below normal or release if above normal. There is a minimum outflow to ensure the downstream river does not dry up, and a non-damaging release so the reservoir does not reach full capacity. Simplified reservoir operating parameters were used based on expert opinion (outlined in Zajac et al., 2017) given lack of availability of global operational release records.

As mentioned in our reply to Anonymous Reviewer #2, we propose to add a new table (Table A below) to accompany Fig. 1 in the resubmitted manuscript that makes it clearer for the reader to find the open access publications outlining the full methodological details of the key components of GloFAS-ERA5.

**Change:** We have added Table A (below), as the new Table 1 with the following sentence in L105-107 in the resubmitted manuscript: "The open access scientific publications and model documentation that describe the full methodological detail for each key component is provided in Table 1 and summarised below". This table includes reference to the full LISFLOOD model documentation (Burek et al., 2013).

**Table A: Scientific papers and model documentation for the key components in the production of GloFAS-ERA5 v2.1 river discharge reanalysis dataset.**

| GloFAS-ERA5 component | Description | Reference |
|---|---|---|
| ERA5 | Global reanalysis dataset using ECMWF Integrated Forecast System (IFS) model cycle 41r2 from 1979 to present | Hersbach et al. (2020) |
| ERA5 runoff | Surface and sub-surface runoff within ERA5 generated using the HTESSEL land surface model | Balsamo et al. (2009) |
| LISFLOOD river discharge | River discharge generated using LISFLOOD hydrological and channel routing model to route runoff into and through the river network and provide groundwater storage. LISFLOOD includes lake, reservoir and human water use routines | Burek et al. (2013) |
| Lakes and reservoirs used in GloFAS | Incorporated 463 lakes and 667 reservoirs into the GloFAS river network | Zajac et al. (2017) |
| Calibration of LISFLOOD used in GloFAS | LISFLOOD was calibrated against daily river discharge from 1287 observation stations worldwide | Hirpa et al. (2018) |

Sect. 4.3: The authors provide monthly performance of this dataset. Such information is very useful, however, it is very difficult to interpret this seasonality, because the results are (probably) a mixture of contributions from both the northern and southern hemispheres. Have the authors made similar analysis for each hemisphere? The authors state "Attribution of such biases in the GloFAS-ERA5 reanalysis is outside the scope of this data paper (L293)", however, practical information on the seasonal performance of this dataset will be very beneficial for potential data users. In my view, the authors should add and show, at least, whether a larger bias ratio observed in the months of November to March than the other months (Fig 7c) is attributable to winter discharge from the northern hemisphere or summer discharge from the southern hemisphere (or a mixture of them; or from some specific regions).

**Response:** This is a very good suggestion. We have conducted your proposed analysis (Fig. A) and found that while the overall GloFAS-ERA5 monthly performance in each hemisphere does not change substantially from the global analysis (Fig. 7 in the original paper), there are some differences worth reporting.

**Change:** Fig. A is added to the resubmitted manuscript as a new Fig. 9 together with the following additional paragraph in Sect. 4.3:

"Results are grouped into northern (n=1268 stations) and southern (n=533 stations) hemispheres in Fig. 9. The overall GloFAS-ERA5 monthly performance in each hemisphere does not change substantially from the global analysis (Fig. 8). Nevertheless, there are some differences. The KGESS and bias ratio from the northern hemisphere (Fig. 9a and c, respectively) tend to follow the global analysis most strongly (i.e. Fig. 8a and c, respectively), which is not surprising given 70 % of all stations are located in the northern hemisphere. However, a higher proportion of southern hemisphere stations show large positive biases from April to June, compared to November to March in the northern hemisphere. The largest proportion of stations with negative KGESS in the southern hemisphere are found from August to October (Fig. 9a). These months correspond with lower southern hemisphere correlation (Fig. 9b) and a higher proportion of stations with large positive variability ratios (i.e. GloFAS-ERA5 has higher variability than observed river discharge)."

[Figure]

**Figure A: Performance metrics for each month by hemisphere. Modified Kling-Gupta Efficiency Skill Score (KGESS) (a) with decomposition of KGE' into Pearson correlation (b), bias ratio (c), and variability ratio (d). Brown (green) boxes represent the IQR and horizontal grey line the median for the northern (southern) hemisphere. Whiskers extend to the most extreme data point, unless the data point is more than 1.5 times the IQR from the box and is instead represented as an outlier (grey diamond).**

MINOR COMMENTS L139:

Is a one-year spin-up enough for this simulation? Probably this depends on the groundwater module or dam operation schemes (the information is not clearly written in the current manuscript, though) used in this model.

**Response:** The initialisation routine within LISFLOOD has been designed so that one year is sufficient to estimate the initial state of all the state variables. For state variables that are fast responding and can reach equilibrium quickly (e.g. storage in the upper groundwater zone), a value of 0 can be used at start of the run and one year is more than long enough. However, for other state variables that are slowly responding it is correct that one year can be too short in some model calibration routines, especially for catchments with large groundwater stores. To avoid the need for very long spin up periods, LISFLOOD calculates a "steady-state" storage amount for the lower groundwater zone during a long-term pre-run, and thus reduces the lower zone's spin up time (Burek et al., 2013).

**Change:** We have added the following sentence into L153-155 in the resubmitted manuscript: "To avoid the need for very long spin up periods, LISFLOOD calculates a "steady-state" storage amount for the lower groundwater zone during a long-term "pre-run", and thus reduces the lower zone's spin up time (Burek et al., 2013)".

L191: The authors used "1801 catchments" here, but this expression might be confusing if there are multiple gauge stations in a large river system. I think dividing this sentence into two parts (and used "1801 stations" in the former one) will be clearer for understanding.

**Change:** We changed "1801 catchments" to "1801 stations" in L207 in the resubmitted manuscript.

Sect. 4.2: The authors discuss the results by using both the bias ratio (beta) and PBIAS, but this might be confusing. For example, "-9%" in L241 is PBIAS, due to its negative value, I guess.

**Response:** The bias term in the decomposition of the KGE' is the bias ratio $\beta$ but can easily be converted to the more widely used percent bias (PBIAS) by $(\beta - 1) \times 100$ (as shown in L205-206 in the submitted manuscript). PBIAS was also used in Lin et al. (2019) when considering what we mean by a "very good" bias error in global hydrological modelling (i.e. ±20 %), mentioned on L235-236. Our original intention was to report the bias in the text in the more widely used PBIAS form. However, your point is valid that it might actually be more confusing to the reader.

**Change:** We now report bias errors as bias ratios and variability errors as variability ratios as per Equations 2 and 3 in the resubmitted manuscript.

**Response to Anonymous Referee #2**

The paper entitled "GloFAS-ERA5 operational global river discharge reanalysis 1979- present" presented by Harrigan et al., describes re-analysis driven global river discharge simulations that are updated in near real time and distributed through the Copernicus Climate Change Service Climate Data Store. Overall, the paper is well written and provides the reader with an overview on the methods used for data production, file formats and the performance of the data set.

Thank you for your positive comments and constructive feedback. Your clarifications have improved the manuscript and made it clearer for the reader.

Given, that this paper is a data-descriptor and neither a model documentation nor a research article there is little to criticize. Nonetheless, some aspects of the paper would benefit from additional information. My main points are summarized below:

1. Terminology: The data product presented is referred to as "reanalysis". Although the runoff data used to drive lisflood stem from a reanalysis, the presented data product is not an integral part of ERA5. In addition, observational discharge data are only used for calibrating lisflood, but are (to my understanding) not assimilated through a state-updating procedure. Given that the term reanalysis is often associated with state updating, I would find a clarification of the chosen terminology helpful in order to avoid confusion about the nature of the presented data set.

**Response:** We use the term reanalysis to mean the optimal combining of in situ and satellite earth system observations together with models to provide consistent spatio-temporal "maps without gaps" of land, ocean and atmospheric variables of interest as per Hersbach et al. (2020) and is now common in Earth System Modelling.

**Change:** Our definition does not change from that reported in the original submitted manuscript on L42-45. However, Hersbach et al. (2020) ERA5 global reanalysis peer-reviewed paper has since been published, so the citation is updated.

2. Transparency of the data production process: Although the paper does a good job in summarizing the workflow resulting in the presented data set, the amount of information presented is not sufficient to replicate the data. While I acknowledge that a description of ERA5 or Lisflood are beyond the scope of the paper, there are a number of essential technical steps that are not described. Open questions include, but are not limited to, (i) how was ERA5 output disaggregated to the finer resolution, (ii) how was lisflood calibrated (are the data used for validation independent of the data used for calibration), (iii) what does it mean that reservoirs are included (e.g. is management also simulated), etc. I realize that some of these questions are also treated in other publications but for a user of the data set a comprehensive overview with more details would be essential to fully understand the capabilities (and limitations) of the data.

**Response:** Your overall point on the need for additional clarity, especially in regards to the hydrological modelling detail, was also raised Anonymous Referee #1. It is indeed a balance within this data paper to focus on the description of the GloFAS-ERA5 dataset and its evaluation, while providing sufficient detail on the modelling methodology to allow users to gain an understanding of the capabilities and limitations. Our intention is to provide only a summary of the modelling methodology that is already described in full detail in the published literature.

**Change:** We have added Table A (below), as the new Table 1 with the following sentence in L105-107 in the resubmitted manuscript: "The open access scientific publications and model documentation that describe the full methodological detail for each key component is provided in Table 1 and summarised below". This table includes reference to the full LISFLOOD model documentation (Burek et al., 2013).

Responses and changes to your individual queries are given below:

i.) **Response:** This question was also asked by Anonymous Referee #1. In order to be consistent with the operational GloFAS procedure, the runoff fields from ERA5 were downscaled using the simple nearest neighbour method from the native resolution to the 0.1° LISFLOOD grid. The task was done using the open source 'pyg2p' module with interpolation 'grib_nearest' option in Python (https://pypi.org/project/pyg2p/). No weights were applied to runoff values and terrain effects within the ERA5 cell were not considered.

**Change:** We have added the following sentence into L151-153 in the resubmitted manuscript: "In order to be consistent with the operational GloFAS procedure, the runoff fields from ERA5 were downscaled using the simple nearest neighbour method from the native ERA5 to the 0.1° GloFAS grid."

ii.) **Response:** The LISFLOOD version used here for GloFAS-ERA5 v2.1 was calibrated by Hirpa et al. (2018) using an evolutionary optimisation algorithm against daily river discharge from 1287 stations worldwide. For each station, the record was split in two for calibration and validation. If the record was shorter than eight years, four years were used for calibration and the remainder for validation. If the record was equal to or longer than eight years, half was used for calibration and half for validation, with the most recent period used for calibration.

iii.) **Response:** Reservoir outflow is calculated with a set of simplified rules depending on their filling level, and balances water recharge if storage is below normal or release if above normal. There is a minimum outflow to ensure the downstream river does not dry up, and a non-damaging release so the reservoir does not reach full capacity.

**Change:** Added Table A as new Table 1 for clarity and added the following sentence to the limitations in L334-336: "simplified reservoir operating parameters were used based on expert opinion (outlined in Zajac et al. (2017)) due to lack of availability of global operational release records".

**Table A: Scientific papers and model documentation for the key components in the production of GloFAS-ERA5 v2.1 river discharge reanalysis dataset.**

| GloFAS-ERA5 component | Description | Reference |
|---|---|---|
| ERA5 | Global reanalysis dataset using ECMWF Integrated Forecast System (IFS) model cycle 41r2 from 1979 to present | Hersbach et al. (2020) |
| ERA5 runoff | Surface and sub-surface runoff within ERA5 generated using the HTESSEL land surface model | Balsamo et al. (2009) |
| LISFLOOD river discharge | River discharge generated using LISFLOOD hydrological and channel routing model to route runoff into and through the river network and provide groundwater storage. LISFLOOD includes lake, reservoir and human water use routines | Burek et al. (2013) |

| Lakes and reservoirs used in GloFAS | Incorporated 463 lakes and 667 reservoirs into the GloFAS river network | Zajac et al. (2017) |
|---|---|---|
| Calibration of LISFLOOD used in GloFAS | LISFLOOD was calibrated against daily river discharge from 1287 observation stations worldwide | Hirpa et al. (2018) |

3. The output variable (discharge) is "Volume rate of water flow, including sediments, . . .". While I acknowledge that this is likely the variable of interest for flood forecasting, I would appreciate if the volume (or mass) of pure H2O could also be made available (if this does not differ significantly, then a statement explaining this might be useful).

**Response:** The definition given in Table 2 is the generic definition of discharge from rivers and streams by the World Meteorological Organisation (WMO) for hydrological products (https://community.wmo.int/activity-areas/wmo-codes/manual-codes/latest-version). It is used across all river discharge products at ECMWF and on the Copernicus Climate Change Service (C3S) Climate Data Store (https://apps.ecmwf.int/codes/grib/param-db?id=240013). Virtually all hydrological models, including GloFAS-ERA5, simulate the volume rate of water only due to inherent simplifications of reality.

4. What is the time resolution of the observations used for validation? I assume daily, but this was not stated explicitly.

**Response:** Yes, the observations are daily, and the evaluation carried out at the daily scale.

**Change:** Now clarified as daily in the abstract (L28), and in Sect. 4 (L194, L198 and L212) in the resubmitted manuscript.

5. Stations used for evaluation come "predominantly" from the GRDC. This is not transparent at all and hinders reproducibility of the study. I assume that some of the data cannot be re-distributed, but an overview (e.g. supplementary table) on the considered stations including some key properties (geolocation, river and station names, data-provider, catchment area, . . .) foster reproducibility of the results.

**Response:** Observations cannot be redistributed by the authors due to licencing agreements but to foster reproducibility of the results we have now included Supplementary Table S1 with the resubmitted manuscript with the following metadata for each of the 1801 stations: *GloFAS_ID, Provider, Provider_ID, Station_Name, River_Name, River_Basin, Country_Name, Catchment_Area_Provider_km2, Catchment_Area_GloFAS_km2, Latitude_Provider, Longitude_Provider, Latitude_GloFAS, Longitude_GloFAS.*

In addition, we include in the Supplementary Table S1 the corresponding performance metrics for each station to allow users to explore the results in more detail: *KGE', KGESS, correlation, bias_ratio, variability_ratio, MAE_mm_per_day.*

**Change:** Supplementary Table S1 included in the resubmission.

6. If there is more than one station per grid-cell only one station is selected. This is OK. However, what is the criterion to select a particular station (random, expert judgment, catchment size,. . .)?

**Response:** When multiple observation stations were matched to the same GloFAS river cell, the station with the longest record was retained. This criterion removed 27 stations from the initial list.

**Change:** Now clarified on L205-205 with bullet point reading: "When multiple observation stations were matched to the same GloFAS river cell, the station with the longest record was retained [27 stations removed]".

7. I personally would find extended global summaries (in addition to medians and IQRs) of the performance metrics useful (e.g. tables with percentiles, or empirical cumulative distribution functions).

**Response:** We have taken your suggestion on board.

**Change:** We now present the performance metrics for all 1801 stations as a cumulative distribution function in the resubmitted manuscript (Fig. A) (inserted as Fig. 4 in the resubmitted manuscript). We also include the performance metrics for each station along with the metadata in a Supplementary Table S1 you suggested in your point number 5. The text in Sect. 4.1 has been modified slightly to incorporate the new Figure.

[Figure]

**Figure A. Cumulative distribution function of performance metrics across all 1801 stations. Modified Kling-Gupta Efficiency (KGE') and Skill Score (KGESS) (a) with decomposition of KGE' into Pearson correlation (b), bias ratio (c), and variability ratio (d). The red dot marks the optimum value for each metric.**

8. The performance assessment focuses predominantly on the skill of the full time series at daily resolution. For some users information focussing on different modes of variability (e.g. seasonal cycle, anomalies of the seasonal cycle, year-to-year fluctuations) would be also of great interest.

**Response:** Thank you for your suggestion. We agree that there are many other exciting potential applications of GloFAS-ERA5 that would be interested in an aggregation of the dataset. But here, focusing on the daily time-step will provide the performance of the dataset at the highest temporal resolution that is of most interest for the vast majority of hydrological applications. We expect and encourage users to undertake their own local evaluation for their specific application as the use of the dataset grows.

9. Accessibility of the data product. I am aware of and support the effort of the Copernicus Climate Change Service but I don't have an account for this at the time being. I am also reluctant to create "random" accounts that I need to keep track of if not really needed. Given the fact that the data are produced by one of the world leading institutions for global weather data (ECMWF) and hare hosted on the Copernicus platform, I assume that the data format will be state of the art.

**Response:** Thank you for your positive comment. A requirement for the GloFAS-ERA5 data to be hosted by the C3S Climate Data Store (CDS) is that state-of-the-art cataloguing, data format (i.e. NetCDF), and standardised metadata and documentation are adhered to: https://cds.climate.copernicus.eu/cdsapp#!/dataset/cems-glofas-historical?tab=overview. This allows GloFAS-ERA5 data to be found through the CDS search catalogue, work with the CDS "Toolbox", and follows the protocol that provides programmatic access to the data via the CDS Application Programming Interface (API). These are valuable tools to allow users to work more easily with large global datasets. More general information on the CDS and how data are delivered can be found here: https://climate.copernicus.eu/climate-data-store.

**Other minor changes to the manuscript during the revision**

1. Alfieri et al. (2019) updated to final published paper, Alfieri et al. (2020)

2. Hersbach et al. (2018) updated to the final published paper, Hersbach et al. (2020)

3. L340-341: Some additional detail on the 'rain bomb' issue in the ERA5 dataset was given in the final Hersbach et al. (2020) paper, and so additional detail on how rare they are (~10 episodes per year) and where they occur (mostly in isolated grid points over orographic areas in Africa) was added to the resubmitted manuscript for the benefit of readers.

We really appreciate the time and insight of both referees in reviewing our manuscript,

Kind regards,
Shaun (on behalf of all co-authors)

**References**

[revised manuscript text omitted]